# Accelerated discovery of perovskite solid solutions through automated materials synthesis and characterization

Mojan Omidvar [1], Hangfeng Zhang[1], Achintha Avin Ihalage [1], Theo Graves Saunders[1], Henry Giddens[1], Michael Forrester [2], Sajad Haq[2] & Yang Hao [1] ✉

Accelerating perovskite solid solution discovery and sustainable synthesis is crucial for addressing challenges in wireless communication and biosensors. However, the vast array of chemical compositions and their dependence on factors such as crystal structure, and sintering temperature require time-consuming manual processes. To overcome these constraints, we introduce an automated materials discovery approach encompassing machine learning (ML) assisted material screening, robotic synthesis, and high-throughput characterization. Our proposed platform for rapid sintering and dielectric analysis streamlines the characterization of perovskites and the discovery of disordered materials. The setup has been successfully validated, demonstrating processing materials within minutes, in stark contrast to conventional procedures that can take hours or days. Following setup validation with established samples, we showcase synthesizing single-phase solid solutions within the barium family, such as $(Ba_xSr_{1-x})CeO_3$, identified through ML-guided chemistry.

The emergence of laboratory automation has brought to light various ways to accelerate each stage of the materials discovery process[1]. Advancements in machine learning (ML), for example, encompass a wide range of activities and have been applied in various areas of the electromagnetics community, including discovering novel composition perovskite materials for photovoltaics[2], optimizing the electronic properties of thin films[3], optimizing copolymers[4], generating databases[5] and so forth[6–10]. In our previous study[11], neural network-derived embeddings effectively identified potential and known perovskites and ferroelectrics. Perovskite structured materials are among the widely researched due to their broad functional properties, which can be utilized in wireless communications, phase shifters, tunable antennas, oscillators and biosensors[12–15]. However, traditional discovery and optimization of perovskite solid solutions are hindered by the extensive chemical diversity and complex processes, that influence their crystal structure and properties. Supplementary Fig. 1 shows the

average time scale for the manual workflow (days-weeks). Given the challenge of obtaining substantial quantities of experimental data, most current ML models rely on computational samples, which limits their accuracy to the quality of the data they are built upon[16]. It has become evident that there is a scarcity of studies that experimentally synthesize and validate materials predicted by ML, highlighting a significant research gap and emphasizing the need for alternative strategies to bridge the gap in experimental materials discovery.

With the maturation of artificial intelligence (AI) and collaborative robots, more "self-driving laboratories" (SDLs) for material discovery are being developed[3,17–23]. Nonetheless, SDLs are predominantly in their early stages of development, and their utilization within the realm of solid solutions remains in its initial phases due to the intricate nature of workflows, difficulties in orchestrating lab devices, the need for multi-objective optimization, and the complexities involved in dielectric measurements for 3D materials. Figure 1 presents a

[1]School of Electronic Engineering and Computer Science, Queen Mary University of London, London, UK. [2]QinetiQ, Cody Technology Park, Farnborough, Hampshire, UK. ✉e-mail: y.hao@qmul.ac.uk

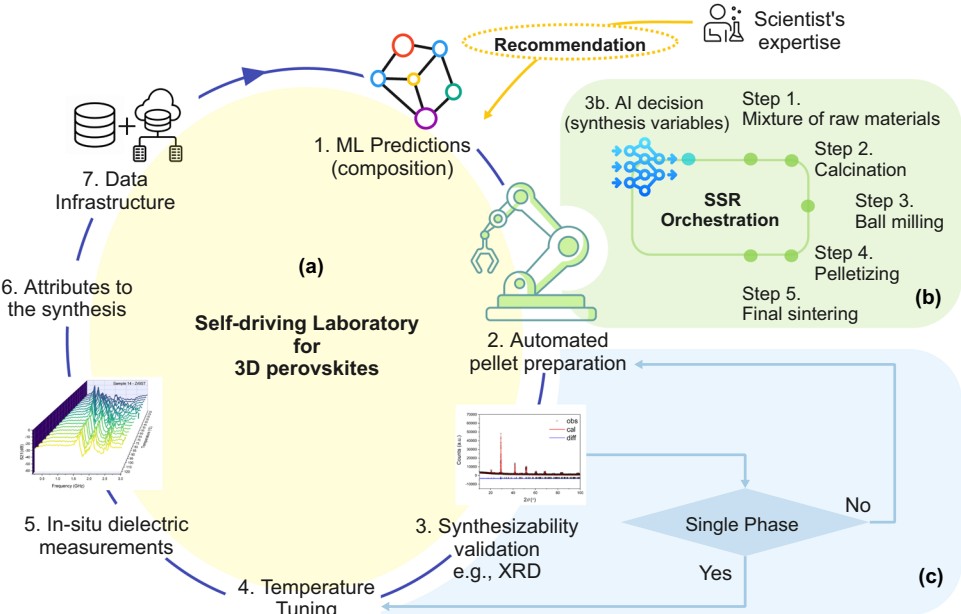

**Fig. 1 | The workflow concept. a** Starting from the top right and moving clockwise, the sequence initiates with machine learning models recommending sample library compositions, followed by automated pellet creation via the solid-state reaction (SSR) method, synthesizability validation (e.g., XRD), temperature tuning, real-time dielectric property measurements, and analysis linking dielectric attributes to the synthesis process. The cycle concludes with the updating of archives and the generation of AI-driven predictions for comprehensive cycle management and decision-making. **b** The SSR workflow encompasses five key steps, expedited by the integration of scientific expertise and AI-enhanced decision-making. **c** The experimental loop validates the initial machine learning predictions.

comprehensive customizable SDL loop for the synthesis and characterization of perovskites, highlighting the process through a combination of both AI and expert scientific input[24,25]. Solid State Reaction (SSR) steps include (Fig. 1b): (1) ball milling, (2) calcining, (3) second ball milling, (4) dry pressing, and (5) sintering[24,25]. Subsequently, material dielectric characterization involves sample preparation such as cutting, grinding and coating conductive electrodes or coplanar waveguide (CPW) transmission lines using sputtering, painting or photolithography[26]. Practical properties for such materials include high tunability ($\tau$) and low loss tangent ($\tan \delta$) at high frequencies. Currently, sample preparation and characterization, even for high-throughput methods, are a predominantly manual process[16,27]. The brittleness of ceramics complicates their handling by robotic arms, often resulting in defects and surface cracks. In addition, high-frequency measurements often require intricate setups and precise calibration for signal integrity and noise reduction. Without effective property characterization, the main benefits of SDLs, such as autonomy and the capacity to independently correlate synthesis with properties, are not fully realized[28].

Previously, exemplar high-throughput experimentation (HTE) and/or laboratory robotic tools have been demonstrated for automating the individual stages in SSR workflows, such as powder dispensing and mixing[29], sintering[30], mechanical testing, and evaluation of structural properties[16]. These individual automated tasks have been reviewed in depth as documented in the referenced literature[16,27]. Vecchio et al. suggested additive manufacturing alloy development followed by structure screening[29]. Sintering conditions in SSR significantly affect phase stability, reaction rates, grain size and density, impacting the dielectric performance of solid solutions[31]. However, conventional sintering has several drawbacks, including long processing time (2–6 h), high energy consumption (at 1200–1800 °C) and $CO_2$ emissions, making it unsuitable for high-throughput experiments[32].

Although alternative sintering techniques like spark plasma sintering (SPS)[30], microwave synthesis[30], flash sintering (FS)[33], and

photonic sintering[34], have been explored to reduce processing time and cost, they come with their own challenges, such as the need for specific equipment or difficulties in experiment reproducibility. Alternatively, CALPHAD computational methods[29], and density functional theory (DFT) calculations[35], have been used for assessing phase stability under various conditions. These approaches often presume an equilibrium thermodynamic state and may not capture the complexity of perovskite oxide, requiring thorough analysis. Furthermore, incorporating temperature effects into ab initio simulations is complex and computationally intensive, often requiring specialized techniques like quasi-random structures or molecular dynamics[36], which are challenging for ML-predicted materials due to data limitations. In addition to synthesis, none of the high-throughput approaches can be directly applied to characterize the dielectric characteristics of bulk ferroelectric materials at high frequencies. Typically, dielectric properties at microwave frequencies are retrieved through s-parameter measurements on transmission lines[37], resonant cavities, and/or free-space techniques, where the propagation characteristics of the Radio Frequency (RF) wave are directly dependent on the properties of the material under test. In this study, we develop an automatic platform designed for rapid sintering processes and high-throughput microwave property measurements, facilitating fast materials screening and characterization in a matter of minutes (Fig. 2). This advancement significantly reduces the time and labor traditionally required for generating validation datasets, streamlining the characterization process. A free-space sensor allows the robot to screen the thermal tunability with no sample surface modification required. Immediate post-sintering measurements eliminate the need for reheating samples and improve ML models by rapidly revealing correlations between processes and structures. Here, we design a central hub with a graphical user interface (GUI) on MATLAB for orchestrating our lab instruments and data management, offering the capability to incorporate additional devices for expanding the workflow in the future. Communication between the process management module and the devices are achieved using various communication protocols (TCP/IP over WIFI/

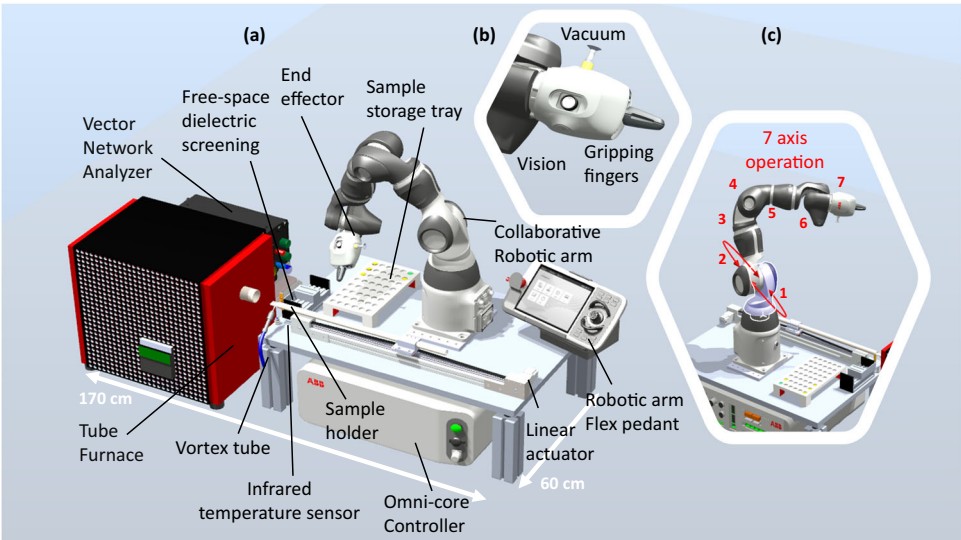

**Fig. 2 | Robotic laboratory for high-throughput synthesis and characterization. a** Digital representation of the automated platform for rapid sintering and free-space dielectric screening at GHz frequencies for solid-state ferroelectrics. The setup explores the relationship between sintering conditions and functional properties. **b** The robotic arm interacts with disc-shaped samples using a vacuum module mounted on the smart end effector, achieving a maximum transfer velocity of 1.5 m/s. **c** The robot used here is a 7-axis collaborative robotic arm-type (ABB single-arm YuMi), programmed based on C language.

LAN, etc.). This enhances the flexibility and modularity of the setup, enabling rapid collection and analysis of data. Supplementary Movie. 1 showcases the experimental setup in action, providing a visual demonstration of the process. Through continuous learning from experimental outcomes, we can downsize datasets, and ML models can propose modifications in composition or processing factors to attain particular desired dielectric properties.

## Results and discussion

In a single cycle that supports the integration of ML modeling, the final step in the SSR synthesis workflow for 3D ferroelectrics (sintering) was linked to high-frequency dielectric characterization in an automated approach (Fig. 3). Each iteration of the automated workflow involved the following: (i) a robotic arm transferred green pellets from the sample tray to the furnace, (ii) samples were subjected to rapid sintering (different times and rates) and (iii) 20-min dielectric analysis during sample cooldown and sample classification by robot (see the methods section for details). We focused on perovskite structured oxides with well-documented data, such as $Ba_{0.6}Sr_{0.4}TiO_3$ (BST)[38–41], and $BaTi_{1-x}Sn_xO_3$ (BTS)[42], providing substantial information on their synthesis and ferroelectric properties for comparison. Additionally, samples from the barium-based perovskite family, such as $Ba_xSr_{1-x}CeO_3$, were selected from ML prediction for comparative analysis. Currently, our research is driven by curiosity, as we systematically filter and categorize ML-screened materials, taking into account their feasibility for synthesis and their potential to unveil optimized properties or uncharted applications. As a first step in validating the methodology, a solid-state reaction method was used to prepare multiple compositions from 3 family groups (Fig. 3.7b): B site (Ta/Nb/Zr/Pure/Hf) doped in BST ($Ba_{0.6}Sr_{0.4}Ti_{0.02}O_3$), 3 BTS ratios ($BaTi_{1-x}Sn_xO_3$) (referred to as BTS12, 14 and 16), and several ML-predicted perovskites. The physical form of all the samples before automated rapid sintering consisted of uniform dielectric discs with a consistent thickness of 1.5 mm and radius of 15 mm.

The results section is structured as follows: (1) Evaluation of the automated rapid sintering process, (2) Observing the influence of rapid sintering on the BTS family group, (3) Evaluation of automated free-space dielectric screening and comparison with four commonly used techniques at both low and high frequencies, (4) Exploring synthesizability of ML-predicted compositions, and (5) Elaboration on the unique dielectric characteristics of the Pure-BST sample after rapid sintering.

## Evaluation of automated rapid sintering

The doped-BST samples are used for evaluation of the automated rapid sintering setup. Doping is an effective strategy to modify the perovskite structure and improve dielectric tuning performance[43]. Density measurements for all 4 BST compositions (Ta/Nb/Zr/Hf) show dense structures (over 90%) when sintered at 1400 °C for 10 min. The values of R-factors obtained from Rietveld refinements for all 4 compositions are small, which indicates a good fit to the experimental XRD (X-Ray diffraction) data (Supplementary Fig. 2). This is consistent with the system, possessing structure as previously reported for BST[25]. No secondary phase indicating impurities or contamination was detected in the diffraction peaks of the doped ceramic samples. From the SEM (Scanning electron microscopy) data, it is noted that the rapid sintering method resulted in a noticeably different grain structure with the presence of both large and very small grains but an average grain size of 10–15 μm, which is close to that observed for the slow rate conventionally sintered sample ($10 ± 2$ μm)[25], and the 1–5% porosity should only have a minor effect. Thus, material performance should be dominated by the effect of the dopant additive. All four BST compositions were sintered in just 60 min, saving time and energy compared to other sintering procedures that can take hours per sample[25]. For BST a fixed sintering time and temperature was chosen based on literature as a starting point. However, annealing optimization can be achieved by adjusting sintering variables.

To illustrate this feature, optimal sintering conditions for the BTS family were investigated via automated rapid sintering with temperatures ranging from 1100 °C to 1400 °C (with 50 °C intervals). As shown in the priority matrix in Fig. 3.5b, the optimal sintering condition is assessed based on sample density, crystal structure, grain morphology and dielectric properties, comparing these factors against those sintered by conventional methods. A total of 7 identical BTS12 pellets were fabricated for testing. BTS is a relatively new material and has the potential for integration into tunable devices due to its nonlinear dielectric permittivity and low loss-tangent at low frequencies[44]. By changing the Sn-doping ratio, the dielectric constant and loss may be

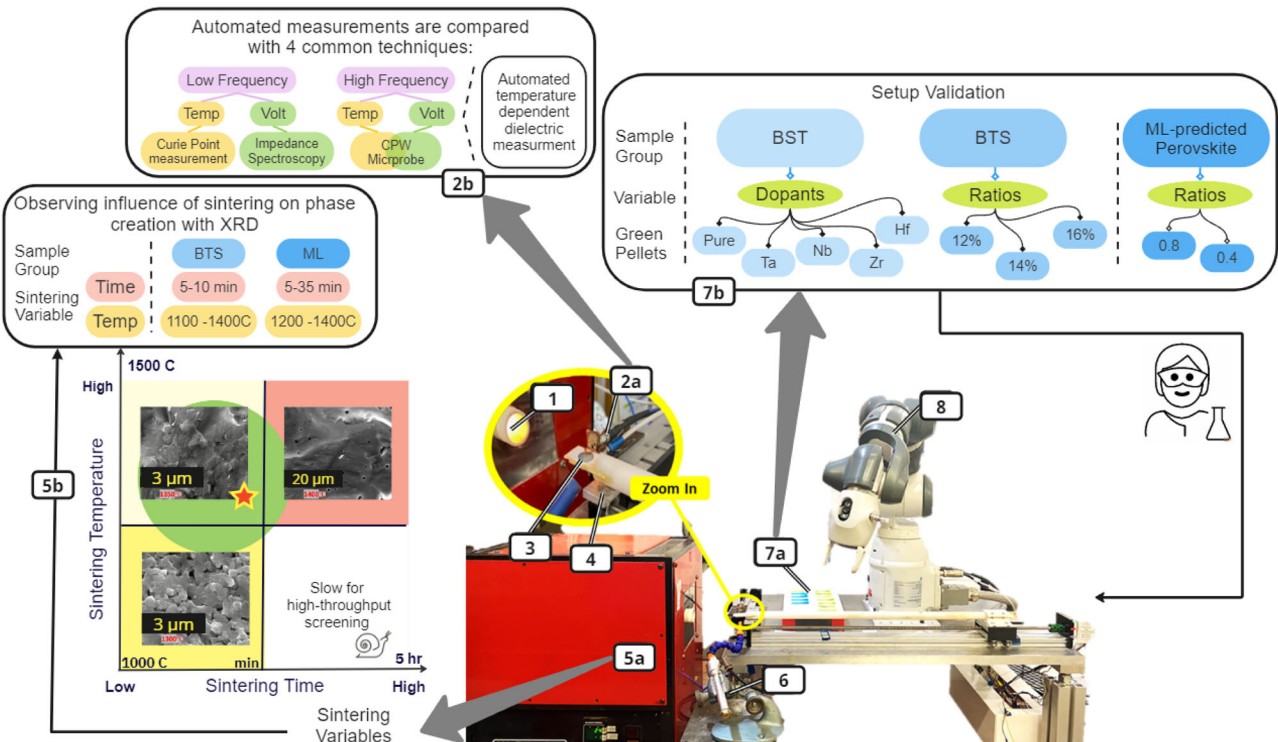

**Fig. 3 | The experimental setup for automated sintering and dielectric characterization of dielectrics.** (1) Tube furnace, (2a) automated dielectric sensor connected to VNA (vector network analyser) and MATLAB graphical user interface, (2b) the four characterization methods used for evaluation of automated dielectric measurements, (3) sample holder and sample under test (SUT), (4) infrared temperature sensor, (5a) automated high-temperature tube furnace, (5b) priority matrix for sintering conditions and variables used for synthesizing single phase BTS and ML-predicted samples (refer to Supplementary Fig. 4a for detailed SEM images The red region indicates long duration and high energy consumption, yellow indicates resulting porous structures and secondary phases, and green indicates conditions that are both energy-efficient and sufficiently fast, (6) vortex tube, (7a) sample tray, (7b) introducing sample groups and green pellets fed into the experimental setup by scientist, and (8) collaborative robotic arm.

adjusted to the desired levels. However, finding the optimum sintering conditions for the BTS family is still an active area of research. The rapid sintering of all 7 samples took a total of 70 min, followed by an additional 20 min per sample for dielectric screening via the automated setup. Supplementary Fig. 3 shows the density of BTS12 ceramics increases with increasing sintering temperature, reaching its maximum values at 1350 °C and 1400 °C. Supplementary Fig. 4a shows SEM data of BTS12 samples sintered at 1300 °C (porous), 1350 °C (dense), and 1400 °C (grain growth), illustrating average grain sizes of 200 nm, 1 μm, and 20 μm, respectively. An average grain size of 1–5 μm is reported in the literature for a dense sample[42]. In Supplementary Fig. 2f, g, the XRD results show that BTS12 sintered at 1350 °C produced well-crystallized samples that closely matched the BTS PDF card and had smaller secondary phase peaks. Current-electric field (I-E) and polarization-electric field (P-E) loops were measured with a ferroelectric hysteresis measurement tester for BTS12-1350 °C and BTS12-1400 °C, as they had the highest densities. Although paraelectric phases are usually considered nonpolar, they may still contain polar nanoclusters[45]. These polar entities can give rise to the presence of two current peaks near 0 V in the I–E loop and to a narrow, nonlinear P–E hysteresis loop (Supplementary Fig. 4b). BTS12-1400 °C showed sharper current peaks and more S-shaped P-E curves, which suggests that this sample would likely exhibit the highest dielectric tunability under an applied electric field. The narrow P–E loops indicate that the polar nanoclusters switch rapidly in an AC field, suggesting that they are very small in size. When characterized by the automated sensor from 25 °C to 65 °C, the dielectric permittivity increases as the sintering temperature increases, which can be attributed to the larger grain size and higher crystallinity when increasing the sintering temperature on the basis of the XRD and SEM results. The dielectric performance of these

samples are discussed in later sections. The highest figure of merit (FOM) was obtained from BTS12-1350 °C. Overall, a sintering temperature of 1350 °C for 10 min was chosen for rapid sintering of the BTS family samples.

## Evaluation of automated dielectric characterization

Following sintering, the robot monitored the dielectric performance of all BST and BTS rapidly sintered samples over the frequency range of 0.2 to 3 GHz with no sample surface modification recorded. These discs function as resonators capable of displaying various TM modes, with resonant frequencies calculated directly using Bessel functions. Given the dependency of resonant frequencies on permittivity, the designated automated sensor ensured consistent positioning of samples, resonant-frequencies measurement through the transmission spectrum and temperature monitoring (refer to the methods section for details). As an example, Fig. 4a shows the shift in the transmission response of ZrBST during 20 min of temperature change from 120 to 20 °C, from which the resonant frequency and permittivity were extracted. The dielectric permittivity of ferroelectrics rises as the temperature approaches the Curie point ($T_c$)[46]. Since $T_c$, which, for the BST and BTS families, is below or at room temperature, here, 25 °C is chosen as the base reference point for tunability calculations for simplicity. Five samples are selected for evaluating the characterization method, including three from the BST family (ZrBST, Pure-BST, and HfBST) and two from the BTS family (BTS12-1350 °C and BTS16-1350 °C). The selected samples were confirmed to be at full densities, ensuring accurate dielectric performance representation without distortion from porosity or secondary phases. Figure 4c shows the gradual decrease in dielectric permittivity with increasing temperature for all five samples. The error bars depict the variation in permittivity for

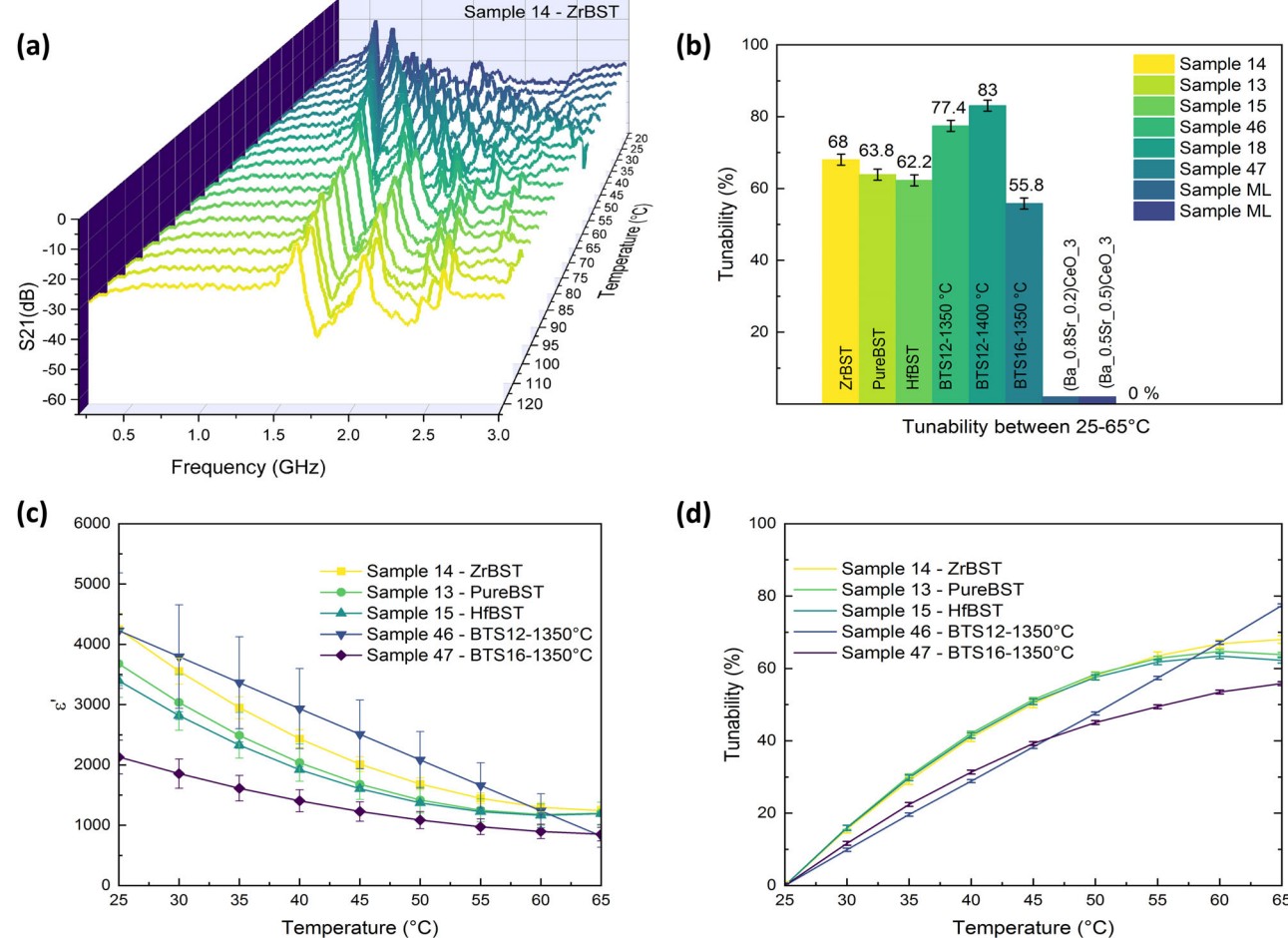

**Fig. 4 | Automated dielectric characterization. a** Automated measurement of the shift in transmission response S21 during temperature change (120–20 °C) for a representative sample (Sample 14 - ZrBST). **b** Thermal variation of dielectric permittivity for selected samples (Error bars represent the permittivity dispersion within the frequency range of 0.6–1.6 GHz). **c** Thermal variation of tunability for five selected samples (Error bars represent the standard deviation associated with linear fittings). **d** Tunability values between 25–65 °C for selected samples (Error bars represent the standard deviation of the system after repeated measurement, refer to Supplementary Fig. 6).

each data point across the frequency spectrum where shifts in resonant frequencies are observed due to temperature changes. These values include the frequency-dependent nature of permittivity observed at room temperature using CPW method, as detailed in the methodology and illustrated in Supplementary Fig. 5. The observed reduction in permittivity from 0.6 to 3 GHz frequencies suggests a relaxation process commonly seen in ferroelectric ceramics[47]. As a result, a correction factor for dispersion has been applied to the tunability calculations. This entailed computing the corresponding permittivity value at ambient for the resonant-frequencies observed at high temperatures. The effect of thermally excited charge carriers on tunability is shown in Fig. 4d. Furthermore, Supplementary Fig. 6 illustrates a close agreement between automated sensor measurements during the cooling and heating of the sample achieving a system error (SD) of only 0.89%. The repeatability of these measurements introduces an error of less than 1.5%. For a detailed uncertainty analysis, please refer to the methodology section. The automated measurements are compared with four commonly used low- and high-frequency techniques. These techniques include impedance analysis and Curie temperature analysis at low frequencies, as well as electrical and temperature tuning via CPW techniques at Microwave frequencies. Here, the term high frequency refers to the range of 0.5 GHz to 5 GHz, while low frequency pertains to the range of 1 kHz to 500 kHz. All electric and temperature tunability measurements are summarized in Table 1.

## Low- and high-frequency measurements

For low-frequency measurements, the variations in the dielectric permittivity and dielectric loss in the frequency range from 1 kHz to 500 kHz were measured under a direct current (DC) bias electric field ranging from 0 to 2 kV mm⁻¹ using an impedance analyzer. The dielectric response results from the short-range rotation of dipoles under the influence of an externally applied electric field[25]. The dielectric permittivity and loss of all five samples gradually decreases when the DC bias field is increasing (Supplementary Fig. 7), illustrating the dielectric tunability shown in Fig. 5a at 100 kHz (the error bars represent the SD calculations for linear fits related to each data point). It is observed that the tunability increases linearly with increased field for all samples. The reason is that the activity of the polar nanoclusters is reduced by applying the DC bias field, resulting in a decrease in their contribution to the dielectric permittivity. Under an electric field of 2 kV mm⁻¹, BTS12 and BTS16 exhibit 58.4% and 38.1% electrical tuning, respectively. For the BST family, ZrBST exhibits the highest electrical tuning (42.3%), which is very close to Pure-BST. Pure-BST shows variations in electrical tuning (13% and 41.1%) and loss responses. To explain this behavior, capacitance-voltage (C–V) measurements are obtained to evaluate the frequency dispersion of this structure. Later in this paper, we elaborate on this particular sample with its distinctive behavior, which we refer to as Pure-BST" (same sample but in its distinct mode). To obtain the real frequency-independent capacitance of this sample, an equivalent circuit model has been proposed.

**Table 1 | All electric and temperature tunability measurements**

| Machine | (a) Impedance spectroscopy (%) | (b) Curie temperature setup (%) | (c) Micro Probe CPW (%) | (d) Micro Probe CPW (%) | (e) Automated Setup (%) |
|---|---|---|---|---|---|
| Frequency | 100 [kHz] | 50 [kHz] | 1.3 [GHz] | 1.3 [GHz] | 0.2–3 [GHz] |
| Field | 2 [kV $mm^{-1}$] | 0 | 4 [kV $mm^{-1}$] | 0 | 0 |
| Temperature | Ambient | 25–65 [°C] | Ambient | 25–65 [°C] | 25–65 [°C] |
| ZrBST | 42.3 (±0.94) | 60.8 (±1.74) | 30.73 (±1.76) | 61.66 (±7.29) | 68.03 (±1.07) |
| PureBST | 13 (±0.79) | 59.9 (±1.65) | 18.92 (±2.06) | 60.38 (±7.53) | 63.85 (±0.57) |
| PureBST" | 41.1 (±0.79) | - | - | - | - |
| HfBST | 39 (±0.89) | 54.2 (±0.8) | 17.34 (±2.1) | 51.84 (±9.16) | 62.25 (±0.74) |
| BTS12-1350 °C | 60.44 (±3.04) | 58.4 (±2.25) | 36.28 (±1.62) | 41.43 (±11.14) | 77.43 (±0.44) |
| BTS16-1350 °C | 39.4 (±0.94) | 38.1 (±1.25) | 16.14 (±2.13) | 42.3 (±10.97) | 55.82 (±0.53) |

Error calculations associated with: (a) SD linear fittings, (b) SD polynomial fittings, (c) system SD error, (d) system SD error, (e) SD dispersion linear fittings.

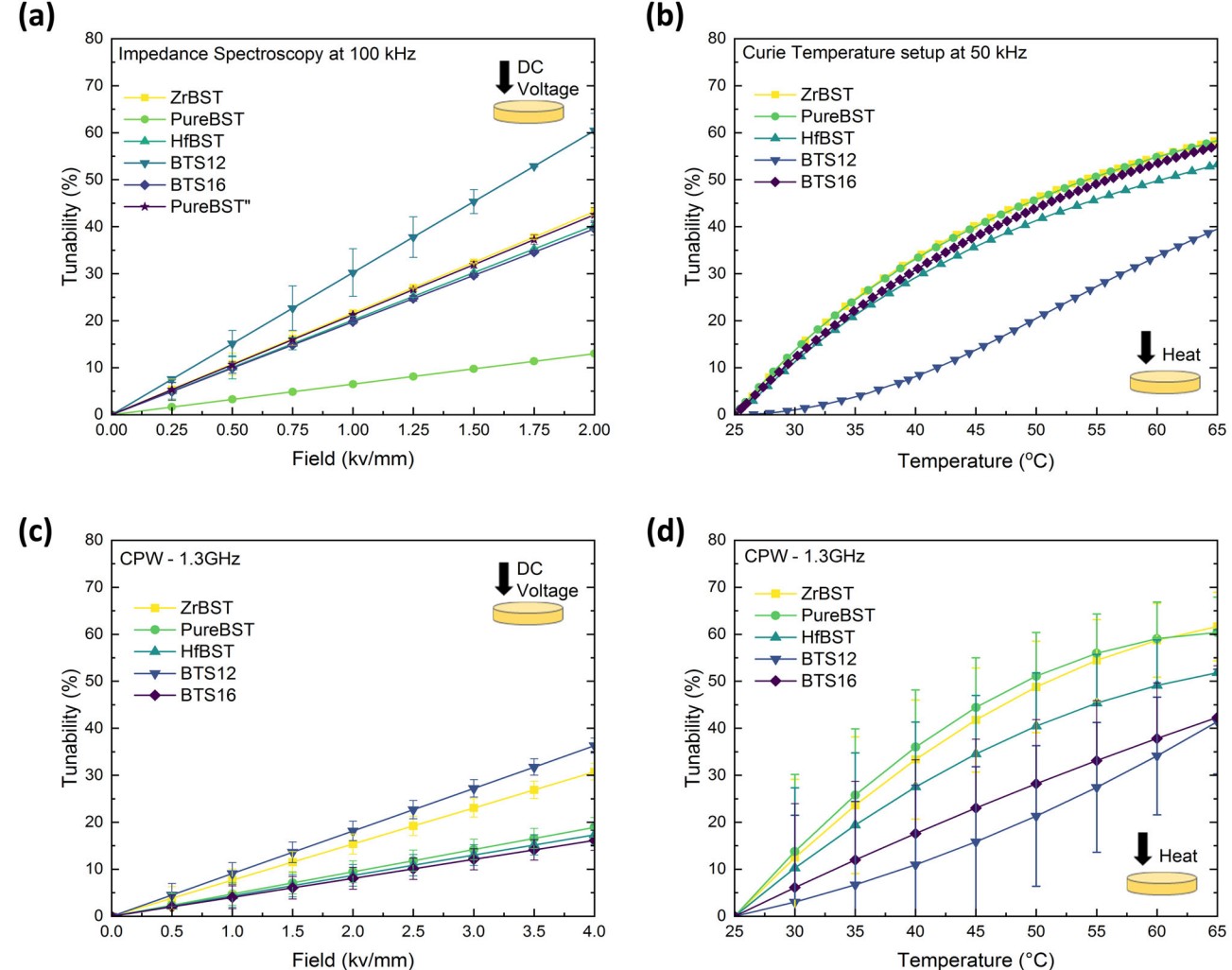

**Fig. 5 | Low- and high-frequency measurements. a** DC-field variation in the dielectric tunability at room temperature for selected samples at 100 kHz (Error bars represent the standard deviation associated with linear fittings), **b** thermal variation in the dielectric tunability (25–65 °C) for five selected samples at 50 kHz (Error bars represent the standard deviation of system after repeated measurement, refer to Fig. S8); **c** MicroProbe CPW (coplanar-wave-guide): DC-field variation in the dielectric tunability at room temperature for five selected samples at 1.3 GHz, (Error bars represent the standard deviation of system after repeated measurement, refer to Fig. S13. **b, d** MicroProbe CPW: thermal variation in the dielectric tunability (25–65 °C) for five selected samples at 1.3 GHz (Error bars represent the standard deviation of the system after repeated measurement, refer to Supplementary Fig. 13. a). Refer to Table 1 for more details.

Calculation via Eq. 2 gives dielectric losses at 100 kHz for the ZrBST, HfBST, BST12 and BTS16 ceramics of 0.013, 0.014, 0.017, and 0.013, respectively. For tunable microwave devices, the FOM is always chosen to represent the tunable performance[48]. For the BST family, when the dopant content is switched from Hf to Zr, FOM values increase from 4.31 to 5.37, yet both are lower than that of Pure-BST", which gives a value of 5.57 (with 41.1% tuning). For the BTS family, the highest tunability and FOM are both obtained for BTS12 due to its high dP/dE

(Supplementary Fig. 3), which results from the active polar nanoclusters inside. Plotted in Fig. 5b are thermal variations of tunability ($T_{min}$ = 25 °C and $T_{max}$ = 80 °C) measured at 50 kHz. Supplementary Fig. 8 illustrates well agreement between measurements during sample cooling and heating. Measurements at lower frequencies are generally considered more reliable due to the technique's simplicity, consistent dielectric response, and higher impedance, minimize loading effects and enhance measurement accuracy, however, they cannot truly reflect material properties at microwave frequencies[49]. For BST family tunability demonstrates a polynomial increase with temperature rise and then stabilizes. While temperature tuning dominates electrical tuning, a 2 kV mm$^{-1}$ bias is comparable to temperature tuning at 47 ± 2 °C. For BTS12, however, a more linear temperature tunability trend is observed, which can be attributed to a shift in the resulting from a change in the doping ratio when compared to BTS16 (as confirmed in Supplementary Fig. 9). The ε of the BTS samples shows a gradual increase with increasing temperature up to $T_c$ and then starts decreasing. A broadened dielectric anomaly appears for BTS16 against temperature, which means that the phase transition from ferroelectric to paraelectric occurs over a wide temperature range. Above $T_c$, the ferroelectric is in its paraelectric phase and does not possess a domain structure but reveals dielectric nonlinearity due to the existence of polar nanoclusters near $T_c$[45]. It can be observed that there is almost 50% more voltage tuning in BTS12 than in BTS16, with only a negligible increase in loss. Tailoring $T_c$ of ferroelectrics is a promising method to enhance tunable performance, as the dipoles inside the polar nanoclusters are easy to rotate under an applied electric field.

For high-frequency measurements (1 to 6 GHz), Coplanar Waveguide transmission line-based measurements under DC bias is used. The CPW pattern in Supplementary Fig. 10 was chosen for this case. Plotted in Fig. 5c, d are the fitted curves of thermal and electrical tunability for the five selected samples at 1.3 GHz. This selection was guided by two key considerations: firstly, it provided a comparable frequency range for analyzing alongside the robot study, particularly as most observed shifts in resonance frequency occurred within the 0.6–1.6 GHz range; and secondly, beyond 2 GHz, complexities such as dispersion and dielectric relaxation become more evident, introducing greater uncertainties. The accuracy of the obtained values depends significantly on the accuracy of the measured S-parameters. Therefore, three different line lengths (0.5 mm, 0.8 mm, and 1 mm) were used. In this way, the calculated characteristic impedance inside was also checked to verify the correctness of the measured values. The tunability data shown in Fig. 5c were obtained by applying DC bias up to 4 kV mm$^{-1}$ through the probe, which resulted in the highest tunability of 30.73% (±1.76) being achieved by ZrBST, among BSTs. Of all BTS samples, BTS12 has the highest tuning percentage of 36.28% (±1.62) with a 4 kV mm$^{-1}$ bias. Compared in Fig. 5d, for all of the BST family samples, when the temperature increases from 25 to 65 °C, tunability follows a polynomial trend. Measurements at both GHz and kHz frequencies in the BTS family indicate that the tunability is less dependent on temperature and highly influenced by the electric field. However, this requires the application of high voltages, which increases the susceptibility to breakdown during operation. It is noted that while both BTS12 and BTS16 exhibit similar temperature tunings at 1.3 GHz, BTS12 demonstrates double the voltage tuning. This discrepancy can be attributed to the modified crystal structure, which shifted its $T_c$ closer to room temperature (Supplementary Fig. 9).

Dielectric properties measurement at such high frequency faces challenges such as increased propagation losses, signal integrity and inevitable noise[50]. In particular, significant uncertainties and errors (referenced in Table 1) arise from probe displacement due to thermal expansion during heating (illustrated in Supplementary Fig. 13). As a result, some research opts to only conduct dielectric property measurements at lower frequencies, using extrapolation methods to predict high-frequency dielectric properties which is used for real applications[51]. This technique, while more practical and cost-efficient, needs further accuracy improvement. The results in Table 1, illustrate the drawbacks of relying only on low-frequency data, as the field tunability of ZrBST at low frequency is comparable to that of Pure-BST, but at high frequency, Zr-BST exhibits higher tuning.

To compare the dielectric performance of robotic versus conventional sintering methods, another batch of BTS12 (BTS12-Con) was prepared by sintering at 1350 °C for 2 h[42]. At 100 kHz frequency, BTS12-1350 and BTS12-Con exhibit dielectric tunability of 60.4% and 79.2% under an electric field of 2 kV mm$^{-1}$, respectively. At 1.3 GHz frequency, the dielectric tunability of BTS12-1350 and BTS12-Con was 36.28% (±1.62) and 30% under an electric field of 4 kV mm$^{-1}$, respectively. Low frequency dielectric tunability, often associated with the coexistence of large ferroelectric domains and small polar nanoclusters, suggests that a higher dielectric tunability indicates a greater concentration of the polar nanoclusters within the system[25]. However, large domains exhibit less activity at microwave frequencies, due to slow dipole response. Hence, at high frequencies, dielectric tunability is primarily determined by the polar nanoclusters[42]. As mentioned in previous sections, these polar nanoclusters are impacted by the sintering condition. BTS12-1350 °C exhibit a higher tunability at high frequencies compared to BTS12-Con. In addition, the dielectric tunability of 36% at GHz frequency is higher than other reported values for BST-based materials, such as 10.5% for BST/Mg$_2$SiO$_4$/MgO[52], and 27% for Mn-BST/MgO[53].

From Table 1, it is crucial to acknowledge that no single method can adequately characterize samples over the entire frequency band; thus comparisons are not intended to yield identical outcomes. Typically, the selection of characterization processes can vary based on the desired frequency, required accuracy, temperature, sample size, contacting/non-contacting methods, and cost. Uncertainty in dielectric measurements is inevitable when characterizing different materials, as noted in previous studies[49,50]. The measurements executed by the robotic platform demonstrate efficiency by reducing analysis time to just 20 min, outperforming traditional, labor-intensive procedures like polishing, applying silver paste, and sputtering CPW pattern, which can extend for hours considering the vast number of ML samples. It also eliminates the need for reheating for dielectric measurements immediately following sintering, thereby minimizing environmental impacts and costs. Furthermore, having a free space technique increases the accuracy of measurements compared to CPW thermal measurements which are highly susceptible to thermal expansion of probes. While the trade-off between speed and accuracy remains a topic of research for high-throughput dielectric screening, results demonstrate a good correlation with both voltage and temperature tuning observed at lower frequencies. It is important to note that rapid data acquisition supports the refinement of our computational models, improving the efficiency of guiding experiments and facilitating an iterative discovery process by correlating synthesis variables with high-frequency functional properties, a step missing in previous high-throughput platforms.

## Exploring single-phase synthesisability of ML-predicted compositions via the automated sintering approach

The automated platform was tested with some selected ML-predicted perovskites. Here, we experimentally validated some compositions from our previous study that introduced an ML-based screening strategy for discovering perovskite solid solutions from a pool of nearly 0.6 million candidates[11]. Two representative samples belonging to the barium family, namely, (Ba$_{0.8}$Sr$_{0.2}$)CeO$_3$ and (Ba$_{0.4}$Sr$_{0.6}$)CeO$_3$ are successfully synthesized in this paper. Briefly, a comprehensive set of potential candidates in the form of (A$_{1-x}$A′$_x$)BO$_3$ and A(B$_{1-x}$B′$_x$)O$_3$ compositions was compiled by exploring various combinations of elements from the periodic table that could occupy the crystallographic A-site and B-site of the perovskite structure. This process

adhered to fundamental chemistry guidelines such as Pauling's valency principle. The ML models were trained on a comprehensive experimental dataset acquired by exhaustively querying the inorganic crystal structure database (ICSD). This dataset consisted of 1758 perovskites and 227 non-perovskites, including their crystal structure information. Due to the dataset imbalance, ML classifiers in our prior work underwent 100 iterations, randomly sampling positive examples (perovskites) to match the number of negative examples (non-perovskites) for training and evaluation. The gradient boosting classifier which produced the best performance with a 94% classification accuracy was used to screen the 0.6 million candidates pool. Materials with a high predicted perovskite likelihood (>0.98) are identified as promising or potentially synthesizable perovskites. This resulted in a list of nearly 5000 compositions, which is still too large for a materials scientist to manually go through and select a few materials for synthesis. Here, we apply data mining and human expert knowledge to further down-select highly promising materials for synthesis. Our data mining strategy involves capturing uncharted materials that are closer to existing experimentally realized materials, retrieved by deriving composition embeddings. We use two embedding methods; (1) embeddings extracted from the latent space of a variational autoencoder (VAE) trained on experimental data, as described in our previous work[11], and (2) composition embeddings constructed by taking the weighted average of element embeddings adopted from Mat2vec[54], and Skipatom[55], approaches. The average Euclidean distance from each uncharted composition to its five experimental neighbors is calculated as reported in our earlier work[11]. To calculate these distances systematically, we obtain $ABO_3$ perovskites with energy above the convex hull below 10 meV$^{-atom}$ from the Materials Project database using the Pymatgen package, resulting in 487 compositions. Such materials are generally considered stable or meta-stable. The average cosine distance from each uncharted disordered composition to its parent $ABO_3$ perovskites is then obtained using Mat2vec and Skipatom composition embeddings. Likewise, these distances are used to identify and rank disordered materials that are closer to existing materials

in latent (or embedding) space. Additionally, we performed further data mining with human feedback, (Fig. 1, human operators guide research objectives, and recommend adjustments to the ML operation) considering the potential synthesis routes, with the availability of resources in the lab and chemical toxicity, specifically to remove potentially hazardous or unstable compositions based on existing literature. The ML-predicted materials that rank highest at the end of this data mining strategy are considered most likely to be synthesized successfully. Likewise, the named components from the barium family were identified as having a high likelihood of being synthesized in a single phase. Moreover, additional samples from the likely and the least likely ML-predicted materials were fabricated and tested. However, XRD results revealed multiple phases in these samples (Supplementary Fig. 14). It's worth noting that while there are numerous studies on ML applied to perovskite-type materials[56,57], very few validate ML predictions with experiments. Two major reasons for this are (1) ML models are usually not trained on failed synthesis attempts (lack of such published data), and (2) ML models are not typically informed about the subsequent complex synthesis process. In addition most ML-material studies focus on "local search," limiting exploration to specific compositions or constrained cation/anion choices. While this method may have a higher success rate within existing composition spaces, it often leads to limited discovery of perovskites with desired properties. In contrast, "global search" explores all possible combinations of periodic table elements for perovskite structures. Our prior study[11], utilized a global search approach, screening promising disordered perovskite oxides from diverse element combinations. The present study demonstrates the limitations of relying solely on ML for experimental discoveries and highlights the importance of data mining, human expertise, and guidance from literature to successfully synthesize unexplored perovskites.

The contour plot in Fig. 6a illustrates the screened sintering condition range and their impact on the XRD secondary phase peak for $(Ba_{0.4}Sr_{0.6})CeO_3$. This encompassed the examination of sintering temperatures ranging from 1250 °C up to 1400 °C and sintering

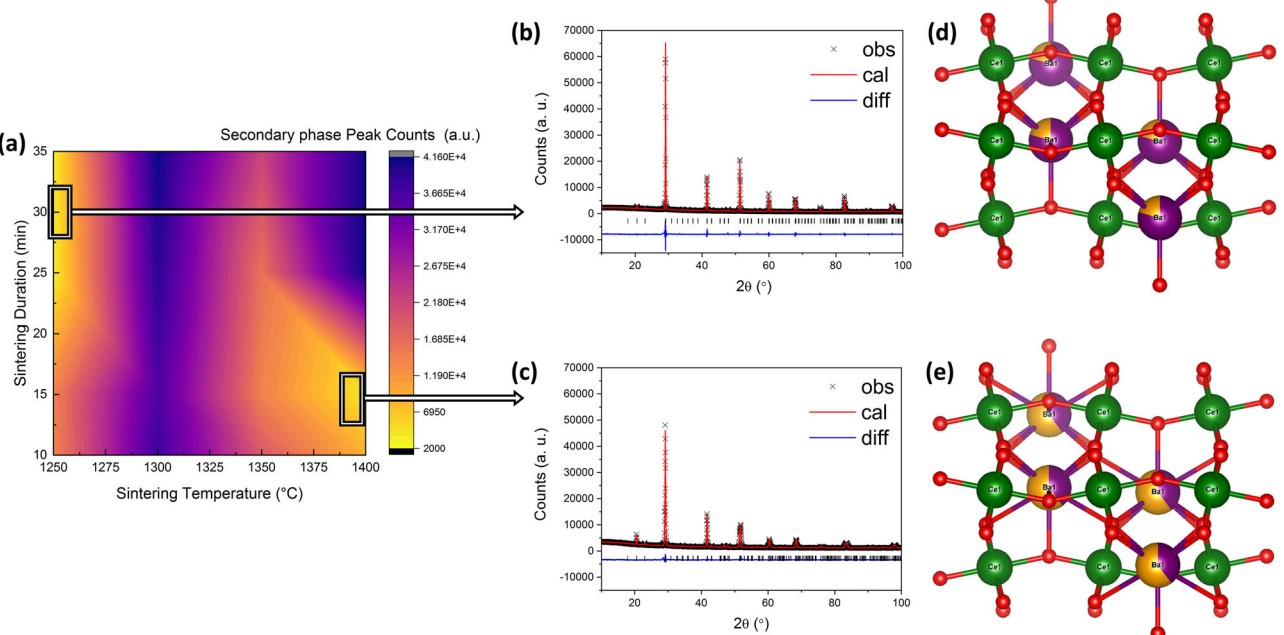

**Fig. 6 | Process-Structure analysis for ML predicted composition. a** The contour plot of automated rapid sintering conditions displays the investigated range of variables and their impact on the XRD secondary phase peak of ML-predicted composition $(Ba_nSr_{1-n})CeO_3$. (The abbreviation "a.u." denotes arbitrary units). **b** XRD profile of single phase $(Ba_{0.8}Sr_{0.2})CeO_3$, **c** XRD profile of single phase $(Ba_{0.4}Sr_{0.6})CeO_3$, **d** crystal structure of $(Ba_{0.8}Sr_{0.2})CeO_3$ (orthorhombic, Pnma), and **e** crystal structure of $(Ba_{0.4}Sr_{0.6})CeO_3$ (orthorhombic, Pnma). (The atom colors represent specific elements: green is Cerium (Ce), red is Oxygen (O), and purple/orange is Barium/Strontium.).

holding times of 10, 15, 25, and 35 min. The contour plot also demonstrates that attaining a single-phase BSC family ML-predicted sample is achievable through two distinct strategies: employing elevated temperatures (>1350 °C) for shorter durations (<20 min) or employing lower temperatures for extended durations. The XRD analysis revealed that $(Ba_{0.8}Sr_{0.2})CeO_3$, which was sintered at 1400 °C for 10 min and $(Ba_{0.6}Sr_{0.4})CeO_3$ sintered at 1250 °C for 25 min, successfully achieved a single phase structure. Figure 6 also shows the crystal structure for both successfully synthesized ML-predicted compositions (Fig. 6d, e). Further details on their cell parameters and density information can be found in Table 2. In an ideal scenario, providing ML models with

experimental recipes would enhance the precision of material synthesis predictions[58]. Fig. 1b illustrates the prospect of incorporating AI decisions, literature data, and scientific expertise in this context. For example, Kirman et al. utilized a deep learning method to characterize perovskite single crystal growth in an HTE cycle, refining experimental conditions based on ML predictions. Unfortunately, generating synthesis predictions for ML-predicted materials remains challenging. With this setup, we generate the initial set of training data, including synthesis and structure data to explore more sophisticated machine learning algorithms that could be integrated into our system to further push the boundaries of innovation.

### Table 2 | Cell parameters and density information for $(Ba_{0.8}Sr_{0.2})CeO_3$ and $(Ba_{0.4}Sr_{0.6})CeO_3$

| Single Phase ML-predicted perovskites | | |
|---|---|---|
| Chemical formula | $(Ba_{0.8}Sr_{0.2})CeO_3$ | $(Ba_{0.4}Sr_{0.6})CeO_3$ |
| Formula weight | 315.505 | 295.621 |
| Crystal system | Orthorhombic | Orthorhombic |
| Space group | *Pnma* | *Pnma* |
| Unit cell dimensions | $a$ = 6.1925(5) Å | $a$ = 6.1692(4) Å |
| | $b$ = 8.7674 (2) Å | $b$ = 8.6751(6) Å |
| | $c$ = 6.1906(5) Å | $c$ = 6.0990(4) Å |
| Volume | 336.10 (1) Å³ | 326.41(4) Å³ |
| Z | 4 | 4 |
| Density(calculated) | 6.235 g/cm³ | 6.016 g/cm³ |
| R-factors | $R_{wp}$ = 0.0628 | $R_{wp}$ = 0.0445 |
| | $R_p$ = 0.0461 | $R_p$ = 0.0335 |
| | $x^2$ = 5.401 | $x^2$ = 2.933 |
| | $R_F^2$ = 0.0626 | $R_F^2$ = 0.0727 |
| Total No. of variables | 24 | 24 |
| No. of profile points | 3649 | 3649 |

### Investigation of the particular dielectric behavior of the Pure-BST" sample

Figure 7a, the SEM image for the Pure-BST sample appears to be covered with irregularly shaped grains of various sizes ranging from very large to very small after rapid sintering. The large grains, which are approximately 10–20 μm in diameter, dominate the image and are scattered throughout. On the other hand, the smaller grains are significantly more abundant, yet they are considerably smaller, varying from 1–5 microns in diameter. These smaller grains are scarcely noticeable at lower magnifications and display a noticeably smoother surface structure compared to the larger grains. The dielectric permittivity and dielectric loss measurements at both low and high frequencies with a DC-bias electric field revealed that this sample structure is not a simple parallel plate capacitor. The dielectric permittivity and loss of the measured ferroelectric cannot be simply calculated from the measured capacitance value and loss of the test structure by using the thickness of the sample and the area of the silver patch. Studies have reported that variations in the thickness of the oxide layer, the density of interface traps, and the bulk doping profiles of high-k gate dielectrics can impact the intrinsic oxide parasitic resistances, series resistances, and interfacial layer[59]. These factors can cause differences between the actual sample structure and the test

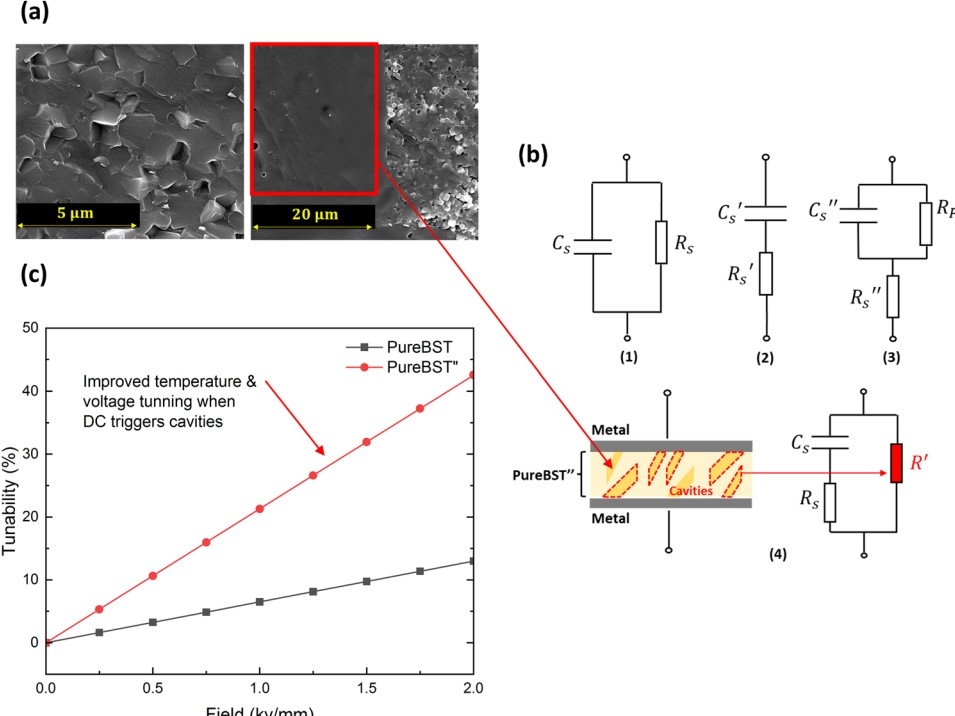

**Fig. 7 | Distinctive behavior of Pure-BST". a** SEM images for the Pure-BST sample rapid sintered at 1400 °C for 10 min. **b** Conventional LCR meters typically measure the device capacitance based on: (1) parallel capacitance model or (2) series, or (3) series and parallel model. (4) The proposed three-element equivalent circuit model for Pure-BST". **c** DC-field variation in the dielectric tunability at room temperature for Pure-BST sample at 100 kHz.

model used by the LCR meter, leading to inaccuracies and difficulties in retrieving the real dielectric constant. The C-V test typically assumes thin oxides to be capacitors with parallel resistors, as shown in Fig. 7b.1. For a high leakage current capacitor with an ultrathin gate dielectric, a three-element circuit model is commonly used (Fig. 7b.2). This design suggests that the measured capacitance during accumulation should not be viewed solely as the oxide layer capacitance but should also take into account the effects of both the gate leakage current and the loss due to AC conductance arising from interface traps and includes all of the series resistances. For low leakage current structures, the device is dominated by a series resistance only, which represents the internal loss (Fig. 7b.3). However, neither of the equivalent circuit methods described above can explain the behavior of the Pure-BST" accurately. The random distributed cavities present in the microstructure of the Pure-BST" causes localized heating of the capacitor regions when a DC bias is applied, leading to the thermal tuning of the samples. Taking into account these conditions and assuming the feature sizes of the structure are much smaller than the wavelength of the microwave signal, the test structure could be analysed following an equivalent circuit with three elements (Fig. 7b.4). The circuit includes an additional parallel resistor that acts as a model for the resistive cavities present in the microstructure of the Pure-BST". This structure possesses peculiar tuning performance with a hybrid and simultaneous thermal/electrical bias as shown in Fig. 7c. As a sanity check, we placed a resistor across the terminals of a capacitor and recorded the same behavior. Although this effect amplifies the dielectric loss, its significance is limited to higher frequencies where only the intrinsic loss becomes relevant. The structure exhibits the capacity to improve its performance through optimization, and one may intend to report on this where careful design and modification of the structure is required.

## Methods

### Automated experimentation

**Automated sintering.** To make a bulk sample, the powder pellets should be sintered into a dense block[60–63]. Typically, sintering for oxide dielectric materials is performed in an air furnace by heating them to the sintering temperature, holding it for an hour or more, and then cooling them to room temperature. However, traditional sintering poses various challenges such as extended processing durations, high energy consumption and CO2 emissions, rendering it impractical for high-throughput experiments. In this work, a compromised rapid heating system based on rapid annealing tube furnaces was developed[64]. This sintering approach offers a controlled and automated method for sintering bulk oxide dielectric materials. Through the utilization of a high-temperature tube furnace model (M.L. Furnaces: ST14/50/180 with a maximum heating temperature of 1400 °C), along with the regulation of the heating and cooling rate, holding temperature, and hold time, the sintering procedure can be fine-tuned to enhance the density and dielectric performance. The use of a Eurotherm controller and an Arduino controller (Arduino Uno) allows for control and monitoring of the sintering process, further enhancing its efficiency and robustness. The use of a robotic arm and a vacuum module gripper (Fig. 2b) allows for the automated transfer of green pellets from the sample tray to the alumina sample holder, which can be inserted and retracted from the tube furnace by a motorized stage.

**High-throughput dielectric characteristics details.** The samples produced through the high-throughput rapid sintering process are uniform in shape, having a consistent thickness and radius. We note from low-frequency measurements and the available literature that the ceramic samples that we are fabricating have high permittivity values (typically >1000) and permeability = 1. As disc-shaped dielectric resonators are known to support resonant modes that depend heavily on the dielectric properties of the material, the automated setup uses a free space dielectric characterization technique to measure the

resonant modes within the hot samples in 0 to 3 GHz frequency range[65]. This is done using the designated sensor which excites and measures the resonant frequency of the $TM_{020}$ mode through closely coupled coaxial loops. The resonant frequency is directly related to the permittivity of the sample through Eq. 1,

$$\varepsilon' = \varepsilon_r = \left( \frac{1}{f_{TM020}} \frac{c}{2\pi} \sqrt{\left[ \left( \frac{5.520}{r} \right)^2 \right]} \right)^2 \tag{1}$$

where $c$ is the speed of light and r is the sample radius. This testing is relatively insensitive to sample geometry but is of limited use in our case, as all the samples have the same geometry. As the samples were removed from the furnace, they were placed between two coaxial loops of the sensor. A non-contact infrared temperature sensor (Optris, CS-LT) was positioned 30 mm below the samples to continuously monitor the temperature of the sample surface. The accuracy of the temperature sensor is ±1.5 °C with data points collected every second. The measuring probe was moved into and out of the measurement position using a motorized stage controlled by the main controller. To increase the cooling rate, a pneumatic solenoid valve was activated, which controlled the flow of compressed air to a cooling vortex tube. The transmission response S21 was measured through a Vector Network Analyser (VNA) model (Anritsu MS2036A). The frequency response was measured at 2-s intervals until the sample cooled to room temperature. The S21 peak shift was detected using MATLAB code, and permittivity was calculated using Eq. 1. To evaluate the performance of the automated measurements, a numerical model representing the sensor was built in CST Microwave Studio. The field maps confirm the excitation of the $TM_{020}$ mode at the corresponding resonant frequency. The resonant frequency was measured and used to calculate the relative permittivity of a sample. The simulated results show good agreement with those values calculated via Eq. 1. The uncertainties for this measurement are calculated by the root sum-of-squares technique (RSS). To use RSS, a set of independent uncertainties were identified such as $\varepsilon$ data fittings, resonant frequency (Half width half maximum (HWHM)) SD errors, and sample diameter measurements.

### Human-in-the-loop analysis

**Sample fabrication.** All samples were prepared by solid-state reaction methods. Stoichiometric amounts of barium carbonate ($BaCO_3$, Aldrich, 99%), strontium carbonate ($SrCO_3$, Aldrich, 99.9%), titanium oxide ($TiO_2$, Aldrich, 99.8%), niobium oxide ($Nb_2O_5$, Alfa Aesar, 99.9%), tantalum oxide (Alfa Aesar, 99.85%), hafnium oxide and zirconium oxide were ball milled at 180 rpm for 20 h. The mixture was dried and sieved, followed by calcination at 1000 °C for 4 h. The powder was ball-milled again, dried and sieved for pelletizing under 150 MPa pressure. For ML-predicted samples, the raw materials were hand-mixed with ethanol by a mortar and pestle. The dried powder was pressed into a pellet with 150 MPa pressure. All samples are pressed into pellets of 15 mm in diameter and ca. 1.5 mm in thickness. As a deviation from the normal process, the samples were partially sintered at 200 °C to enhance their mechanical strength and ensure that they could endure automated handling better.

**Density, phase, and structural analysis.** The effectiveness of the high-throughput rapid sintering system was determined by rapidly sintering samples and subjecting them to various tests, including density measurements, SEM, and XRD. The Archimedes method was used to determine the volume density of the samples. The surface and morphology of the ceramic samples were observed by SEM. The phases present were analysed by XRD. Structural refinement was performed between the experimental and observed XRD patterns. R values ($R_{exp}$, $R_{Bragg}$, $R_{wp}$, $R_p$, $R_f$, $x^2$) are reliability factors to examine the quality of fitting between the experimental and Rietveld-calculated diffraction

peaks. These factors have been reported previously[66]. This assured the accurate representation of material properties in the samples' dielectric performance, preventing distortion due to porosity or the presence of secondary phases.

**Low- and medium-frequency measurement procedures.** For all dielectric measurements below 1 MHz, the top and bottom surfaces of pellets were polished, coated with silver paste (Gwent Electronic Materials Ltd. Pontypool, U.K.) and heated at 250 °C to form electrodes.

At low to medium frequencies (1 kHz to 1 MHz), the dielectric permittivity and loss were measured using a precision impedance analyzer (Agilent 4294 A) under DC bias electric fields (at 0 and 4 kV mm$^{-1}$ at room temperature. The precision of signal frequencies ranging between 20 Hz and 1 MHz is ±0.01%. The dielectric loss tangent value is calculated as Eq. 2,

$$\tan \delta = \frac{\varepsilon''}{\varepsilon'} \qquad (2)$$

where $\varepsilon'$ and $\varepsilon''$ is the real and imaginary part of complex dielectric permittivity[67]. The setup consisted of an auto-balancing bridge that effectively adjusted the values of the internal components in a bridge configuration (Wheatstone), allowing for the measurement of the unknown sample. The following equation was then used to compute the electrical tuning of the samples,

$$\tau(\%) = \frac{\varepsilon_r(0) - \varepsilon_r(E)}{\varepsilon_r(0)} \times 100 \qquad (3)$$

where $\varepsilon_r(0)$ and $\varepsilon_r(E)$ are the relative permittivity at 0 electric field and with an electric field of E applied, respectively. To assess the overall dielectric tunable properties, the figure of merit (FOM) is defined as:

$$FOM = \frac{\text{Tunability}}{\tan \delta} \qquad (4)$$

Additionally, the temperature-dependent dielectric permittivity was measured using a Curie point setup with an LCR meter model (Agilent 4284 A) at 1–100 kHz, which was connected to a cryogenic chamber that controlled the temperature from 80 °C to room temperature.

Current-electric field (I-E) and polarization-electric field (P-E) loops were measured with a ferroelectric hysteresis measurement tester (NPL, UK) with voltage applied in a triangular waveform at 10 Hz. This technique applied very high electric fields (kV mm$^{-1}$) at low frequencies (<1 kHz). Unlike other methods, permittivity was not measured separately from electrical biasing, as the exciting voltage was large enough and slow enough to serve as a biasing field itself. This allowed much higher fields to be applied at the expense of a much lower frequency of testing and ensured that we were not underestimating samples that had higher tunability but needed greater fields to achieve saturation. Thus, permittivity was extracted from the gradient of the P-E loop as a function of the field. All of these measurements were conducted at room temperature.

High-frequency measurement procedure: For high-frequency measurements (range of 1–10 GHz), samples required smooth, flat surfaces and a coplanar waveguide pattern deposited on top. The CPW was achieved by using stainless steel (magnetic grade 400 series) metal masks that were laser etched using fiber laser technology (PrMat) (Supplementary Fig. 10). Samples were placed on a magnetic base, and the masks were positioned on top. They were coated four times with a silver target using a magnetron sputter coater (AGAR auto sputter coater) for 300 s at 40 mA. The permittivity of CPW patterned samples at 1–10 GHz was measured using a VNA along with a

microprobe station (model: EVERBEING INT'L CORP, C-6 Probe Station) and a bias tee. With the EB-050 micro-positioner, the sample location under the probe was precisely adjusted in increments as small as 1 µm. Additionally, the equipment featured a hard chrome-plated platen, vacuum chuck, and standard anti-vibration mounts. Before measuring the samples, a standard calibration substrate (CS-5, GGB Industries) was used to calibrate the probe, and the calibration process was conducted through the short-open-load-through (SOLT) method.

A fixed power exciting wave was passed through the bias tee, the microprobe, one end of the coplanar waveguide, and back out through the other side of the microprobe, a DC block, and back into the second port of the VNA. The transmitted signal S21 was then measured and compared to the reflected signal (S11) to evaluate the permittivity and the loss of the material. The bias tee allowed a DC voltage (0–2 kV mm$^{-1}$) to be applied between the center conductor and the grounds on either side. The S21 phase shift is directly proportional to the permittivity of the CPW substrate, and any shift in the S21 phase during voltage/temperature biasing is due to tuning in the material[68]. Tunability was then calculated through Eq. 3. Shown in Supplementary Fig. 13b, the SD error of the system, related to measurements before and after applying bias, corresponds to an average error of ±1.94%. Additionally, the thermal variation of permittivity (from ambient to 80 °C) was obtained by using a ceramic heater and a low-mass thermocouple. This measurement is susceptible to errors correlated with probes shifting at the microscale due to thermal expansion on heating. This phenomenon can introduce inaccuracies, which are difficult to address, especially at microwave frequencies where precise probe positioning is critical for reliable measurements. Supplementary Fig. 13 compares measurements at room temperature before and after heating, which were used to determine the error values of Table 1.

## Data availability
All data needed to evaluate the conclusions of this study along with the tested training datasets, algorithms/ codes, testing/ validation data, and source data underlying all the figures and tables in the manuscript and the Supplementary Information are provided in the paper, the Supplementary Information and/or the Mendeley Respiratory available publicly at DOI link: https://doi.org/10.17632/j59wsrzt6f.1[69].

## Code availability
The code required to reproduce the high throughput characterization, train the models, and reproduce the ML-predicted materials synthesized in this study are publicly available via the data respiratory link.

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

## Acknowledgements

This work was supported by the "Software Defined Materials for Dynamic Control of Electromagnetic Waves (ANIMATE)" Project (QinetiQ IRAD No. 41025673 and EPSRC Grant No. EP/R035393/1) received by Y.H., and the authors acknowledge QinetiQ and Engineering and Physical Sciences Research Council (EPSRC).

## Author contributions

M.O. and H.Z. contributed equally to this work. M.O. worked on design, implementation, data collection and analysis. Both H.Z. and T.G.S. contributed to material synthesis and characterization. A.A.I. contributed to data mining and machine learning aspects. H.G. provided high-frequency characterization methodology. Y.H. initiated the idea and supervised, directed and coordinated the research. All authors, including M.F. and S.H. contributed to the manuscript editing.

## Competing interests

The authors declare no competing interests.
