## [Transparent Peer Review file · Nature Communications]

Accelerated Discovery of Perovskite Solid Solutions through Automated Materials Synthesis and Characterization

Corresponding Author: Professor Yang Hao

Version 0:

Reviewer comments:

Reviewer #1

(Remarks to the Author)

The authors address the challenges in accelerating the discovery and sustainable synthesis of perovskite solid solutions. The acceleration is particularly discussed in the context of the complexity arising from the vast range of chemical compositions and certain key descriptor such as crystal structure and doping which usually entails extensive manual work. An innovative automated materials discovery approach is proposed by the authors to overcome those challenges. Their approach to expedite the exploration of perovskite materials combines machine learning (ML) for material screening with robotic synthesis, and high-throughput characterization. The authors claim that the proposed self-driving lab is efficient in processing materials in a fraction of time compared to traditional procedures that often may take many hours or days.

In general, the proposed materials discovery and acceleration strategy is innovative and novel. It contains materials and novel approaches that make it suitable for the journal and its wide readership. It clearly demonstrates multi-disciplinary and multi-functional efforts, which are required for well integration of the materials screening output with robotic-controlled synthesis and support from on-line or off-line characterization tools.

To my opinion, there are a few aspects in the methodology, workflow and overall acceleration approach that still need further clarifications to assess and better demonstrate the novelty of the proposed approach. In its current form, the manuscript lacks enough details to judge whether the approach will go beyond state of the art within the ecosystem of digital discovery and self-driving robotic labs.

Introduction:

- How do factors such as crystal structure, temperature, and doping contribute to the complexity of perovskite solid solution synthesis, and why are manual processes time-consuming? What are the key descriptors or approaches that manual work is currently taking to optimize the synthesis process? If those are un-known, how the ML is helping in that regard?
- Related to the above, how do factors such as crystal structure, temperature, and doping contribute to the complexity of perovskite solid solution synthesis, and why are manual processes time-consuming?
- How do the results of the paper contribute to the broader field of materials science and the development of advanced materials for applications like wireless communication and biosensors?
- I am not yet convinced why accelerating perovskite solid solution discovery considered very crucial? How about the economics of the current process and scale up compared to the new process?
- What components make up the proposed automated materials discovery approach, and how do they contribute to overcoming the constraints of traditional methods?

ML model

- It is not quite clear in what ways the automated platform exhibit versatility, and what role does ML-guided chemistry play in the synthesis of single-phase novel perovskite solid solutions? There is a no or little mention and explanation of the machine learning model and its core search algorithm. The author stated that previous ML-based studies for perovskite materials lacked validation with experiments, thus making this work different from those studies. Since no clear process related to training datasets, algorithms and testing/validation is provided, it is hard to judge if the current work offers any new insights. To my opinion the parameter space governing sintering conditions (temperature, time, and cooling rate) is not well articulated in view of workflow optimization and ML-assisted loop.

Methodology/results:

- Some figures were missing or were mis-placed in the original submission. This needs to be checked again in the new revised submission as some references to the figures still do not make sense (captions for Figures 5 and 6 for example and

their references in the text)

- Dielectric Permittivity Changes: How precisely were the changes in dielectric permittivity measured in response to variations in electric field and temperature?

Need to elaborate on the specific mechanisms through which metastable domains contribute to the observed large changes in dielectric permittivity?

- Annealing Conditions: What specific annealing conditions were optimized to induce the presence of metastable domains? How were these annealing conditions determined to be optimal, and what criteria were used to evaluate their effectiveness? Again this could be due to lack of enough information on ML and optimization framework.

- Sintering Effects: How does the sintering process, including phenomena like dopant movement, vacancy formation, and grain growth, directly impact the microstructure of the material?

- Highly beneficial to provide insights into the relationship between the observed microstructure changes and the resulting properties of the material? Is there any descriptor already identified and used in the ML model for prediction purposes?

- Time Frame for Synthesis and Optimization: What are the challenges associated with the time frame for manual synthesis, discovery, and optimization methods, particularly in the context of the exponentially increasing number of conditions and ML-predicted compositions?

Finally: How do the results of the paper contribute to the broader field of materials science and the development of advanced materials for applications like wireless communication and biosensors?

Reviewer #2

(Remarks to the Author)

The creation of new automated workflows for technologically relevant materials is of interest and is timely. The work presented here contains an interesting automated platform, designed for a specific task (making development tractable). Better benchmarking would enhance the value of this work.

The evaluation of the sintering using XRD and SEM is telling. The XRD patterns look good, and the samples are the intended targets with limited impurity phases. The 'odd grains' shown in Figure S2a reveal potential problems. The authors note the presence of the odd grains at 1400, while the progress is clearly present at 1350. Did the samples melt?

The authors note the importance of experimental verification of proposed hypothetical compounds or synthetic routes. I would urge caution regarding the value in computational materials discovery. It should be noted that relative energies of hypothetical materials are often used as a proxy for synthesizability. The complexity of the actual syntheses is correctly noted to be potentially complicated. Having said that, the authors need to better cite the work they are discussing. The literature is rich.

The generation of the two proposed phases using ML techniques is opaque. The authors reference past efforts (with citation), but the phrase "further data mining with human feedback" needs actual information. How were the target phases determined to be "having a high likelihood of being synthesized in a single 297 phase"? How many targets were generated? What selection criteria were used? Were attempts to synthesize and study more than two performed? If so, what were the others and what success rates were observed? The section on the ML efforts does not fit with the rest of the work presented here. The rest is largely a benchmarking study of the automated platform, rightly using established materials. The proposed phases currently read as 'we can study other things too' with two examples provided.

The utility of the robotic platform seems based upon the ability to add and remove samples from the furnace. How does the use of a motorized stage affect cooling rates? How does the rate using this automated platform compared to purely human efforts?

Reviewer #3

(Remarks to the Author)

The paper 'Accelerated Discovery of Perovskite Solid Solutions through Automated Materials Synthesis and Characterisation' presents a new automated laboratory set-up that is designed to accelerate the discovery of new solid state oxide materials, in this instance in the space of perovskite solid solutions. This approach combines machine learning to predict interesting new compounds, robotic synthesis to explore the processing condition space and high-throughput characterisation to assess the newly generated materials. I think that overall this is a timely addition to a very important emerging topic in chemistry, i.e. automated synthesis. There has not been very many demonstrations of solid-state laboratory automation (with the notable exception of the recent A-Lab paper in Nature) and the techniques demonstrated here can make a significant contribution to the field. I do however have some points that I think need to be addressed before this paper is suitable for publication in Nature Communications.

Data: I don't see any indication of where the data used or generated in this study is being made available. In my opinion one of the major benefits of this kind of study is the high-quality data (with good meta data) generated. To make this an impactful study the data should be made publicly available.

In the section on 'Evaluation of automated rapid sintering':

- the authors conjecture that the different grain structure resulting from rapid sintering should not affect the dielectric properties. I think that this needs to be confirmed empirically. This could be achieved by synthesising the same material composition using a more standard slower route and comparing the dielectric performance. This control test is important for the premise of the paper.

- The authors settle on BTS12-1350 as the optimal combination of composition and processing. How does the dielectric performance of this material compare to current state-of-the-art BTS?
In the section 'Evaluation of automated dielectric characterisation'
- The reported results seem to have a very large variation, depending on the technique used. These variations should be discussed. What are the implications for real-world performance?
In the section 'Exploring single-phase synthesizability ...':
- The authors say that certain materials were 'identified as having a high likelihood of being synthesised in a single phase'. There needs to be more detail on how this identification was performed.

Overall assessment

I think that this is a very interesting paper and the results contribute to an important area of materials science. However, I think that there are details missing in the manuscript. With the addition of these details and demonstrating how well newly developed materials compare to the current state of the art, I think that this will be a good paper for Nature Communications.

Reviewer #4

(Remarks to the Author)

In this work, the authors developed an advanced material discovery methodology that integrates machine learning techniques, robotic-controlled synthesis, and high-throughput characterization. To solve the challenge of measuring dielectric properties under electric field, a free-space sensor is developed for online screening of thermal tunability with no sample surface modification required. However, for the current version of this manuscript, the novelty of this is insufficient and might be suitable for other journals. The following issues should be addressed:

1. In the initial validation step, the authors chose multiple compositions from 3 family groups. The rationale behind this selection needs further elaboration. How do the authors ascertain the ferroelectric properties of a material prior to its synthesis?
2. The authors noted that the sintering temperature directly influences the grain sizes of the samples. Would the sintering temperature similarly have a direct impact on the dielectric characteristics of bulk ferroelectric materials?
3. Thermodynamic phase diagrams have proven to be successful predictive tools in comprehending scale growth (J. Phys. Chem. Lett. 2022, 13, 6236–6243). The authors are encouraged to delve into a discussion on the formation of secondary phases from a thermodynamic perspective.
4. The authors are advised to provide a detailed introduction to the methodology of machine learning techniques and DFT calculations within the manuscript, rather than solely referencing their prior work. Moreover, the performance and discussion of ML should be added in the manuscript.
5. The machine learning model employed in this study focuses solely on the synthesis feasibility of perovskites. In order to enhance the exploration of functional materials possessing both high synthesis feasibility and exceptional properties, could the authors contemplate developing a machine learning model that takes into account both synthesis feasibility and ferroelectric properties?

Author Rebuttal letter:

Response to reviewer 1

Reviewer #1 (Remarks to the Author):

The authors address the challenges in accelerating the discovery and sustainable synthesis of perovskite solid solutions. The acceleration is particularly discussed in the context of the complexity arising from the vast range of chemical compositions and certain key descriptor such as crystal structure and doping which usually entails extensive manual work. An innovative automated materials discovery approach is proposed by the authors to overcome those challenges. Their approach to expedite the exploration of perovskite materials combines machine learning (ML) for material screening with robotic synthesis, and high-throughput characterization. The authors claim that the proposed self-driving lab is efficient in processing materials in a fraction of time compared to traditional procedures that often may take many hours or days.

In general, the proposed materials discovery and acceleration strategy is innovative and novel. It contains materials and novel approaches that make it suitable for the journal and its wide readership. It clearly demonstrates multi-disciplinary and multi-functional efforts, which are required for well integration of the materials screening output with robotic-controlled synthesis and support from on-line or off-line characterization tools.

To my opinion, there are a few aspects in the methodology, workflow and overall acceleration approach that still need further clarifications to assess and better demonstrate the novelty of the proposed approach. In its current form, the manuscript lacks enough details to judge whether the approach will go beyond state of the art within the ecosystem of digital discovery and self-driving robotic labs.

We sincerely appreciate the time and dedication invested by reviewer in providing thoughtful and valuable feedback on our manuscript.

Introduction:

1. How do factors such as crystal structure, temperature, and doping contribute to the complexity of perovskite solid solution synthesis? why are manual processes time-consuming? What are the key descriptors or approaches that manual work is currently taking to optimize the synthesis process? If those are un-known, how the ML is helping in that?

Re: The synthesis of perovskite solid solutions is a complex process, which can be influenced by various factors. Crystal structure, temperature and doping play significant role in the materials preparation, which influence the materials's electrical properties. The crystal structure can vary based on the compositions, which affect functional properties of the material, such as ferroelectricity, conductivity and magnetic property. Sintering temperature is the main influential step in the synthesis of ceramics which affect the crystallization rate, phase stability, grain size, density and the quality of final ceramics. Doping is a chemical modification to introduce elements into the perovskite structure to tailor its crystal structure and physical properties. Each individual task involves adjusting the stoichiometry from 1 to x conditions, resulting in a vast matrix of possible synthesis combinations. This limitation is further escalated due to the need for the exploration of a virtually infinite numbers of compositions and therefore trial-error based discovery, and optimization methods are extremely time consuming. Conventional approaches to discovering and refining materials, including perovskite solid solutions, are often laborious and resource-intensive. This is attributed to the vast chemical space, repetitive tasks, the need for manual sample modification before testing, and the reliance on the skill level and expertise of technicians. Additionally, optimizing traditional tools and methods for the characterization to achieve both speed and efficiency can be challenging. In particular, material sintering is one of the most time consuming steps in the synthesis workflow (typically 12 hours per sample including heating and cooling steps). Another bottleneck is the challenges associated with high frequency characterisation. In addition every material discovery workflow can be unique to its hypothesis question and requires specific

1

adjustments to previous automated setups. We have now modified the manuscript accordingly on Page 2, Ln 35-39, Page 3, Ln 58-64, and Page 3, Ln 66-90.

The key descriptors or approaches in manual works are based on systematic adjustment to: find the optimal stoichiometric ratio of the constituent elements of existing materials, doping with certain elements which can affect the physical properties, improving synthesis tasks such as ball milling methods which are employed to enhance the homogeneity of the powder, improvements in sintering conditions including temperature and duration, to achieve dense, well-crystallized samples with desired properties. One way is proposed by Dreger et al. such that an approach centred around establishing a novel data model based on ontologies to form the cornerstone of a remarkably adaptable data infrastructure tailored for fabrication, measurements, and simulations. However, incorporating different variables, especially, the temperature in ab-initio simulations may not be straightforward and very computational demanding especially for compositionally disordered materials such as perovskite solid solutions. We have been exploring the idea of ML for material discovery over the past few years. One possible avenue which we explored is representing new potential perovskites and existing experimental ferroelectrics using embedding vectors extracted from a neural network and identifying new materials that are closer in distance to known ferroelectrics. While this may improve our chances of discovering new materials with functional properties, we realized that there may be issues with synthesisability of these new materials, therefore, we have designed this setup for high throughput experimentation and validation.

In this study, our ML focussed on global search where we attempt to experimentally discover new materials from nearly all possible candidates that can exist in a quaternary disordered format, with the hope of discovering materials with novel properties. Another idea that we examined over time is the so called local search which may yield in higher success rate in synthesising novel materials with target properties, such as ferroelectricity. With this, the ML screening is applied to a limited set of candidate materials, for example, those that contain A-site and B-site elements from a widely known family of compositions that are likely to form stable ferroelectrics, but with properties not very different from existing ferroelectrics in the same family. For instance, Balachandran et al., explored $x\text{Bi}[\text{Me}^2_y\text{Me}^3_{(1-y)}]\text{O}_3(1-x)\text{PbTiO}_3$ -based ferroelectric perovskites using ML, where the choice of Me^2 and Me^3 come from a small set of elements. [Balachandran, P. V., Kowalski, B., Sehrioglu, A. & Lookman, T. Experimental search for high-temperature ferroelectric perovskites guided by two-step machine learning. Nat Commun 9, 1668 (2018).] According to their study, there are over 150 compositions already studied in this family which indicates higher likelihood of synthesis of altered, but not yet synthesised materials. In fact, this study demonstrates several successful syntheses of new materials. This is what we refer to as local search.

2- Related to the above, how do factors such as crystal structure, temperature, and doping contribute to the complexity of perovskite solid solution synthesis, and why are manual processes time-consuming?

Re: This appears to be the same as question 1.

3- How do the results of the paper contribute to the broader field of materials science and the development of advanced materials for applications like wireless communication and biosensors?

Re: In this manuscript, the high-throughput materials preparation and characterization at microwave frequency was performed and results were analysed and screened for wireless communication and

biosensors. For instance, the changing of the dielectric permittivity with temperature indicate the tunability performance of perovskite materials which is an important indicator for microwave communication applications. Perovskite materials can be modified to exhibit specific responses to biological stimuli, making them suitable for biosensors. By modifying the synthesis method we can enhance sensitivity, selectivity, and stability of biosensors, which are crucial for medical diagnostics,

2

environmental monitoring, and biosecurity. From a materials perspective, our research successfully synthesizes two novel ML-predicted perovskites in the barium based perovskite family. Our automatic rapid sintering platform can be used for a broader material range, particularly for those oxide materials which need sintering at high temperature being x12 faster than conventional methods. Furthermore, when compared the dielectric performance of rapid sintered sample with its identical twin prepared using conventional methods, we observed optimised tuning performance at GHz frequencies (Please see Page 13, Ln 299-304). In our study, we also uncovered unique dual tuning behaviour for Pure BST as a result of the rapid sintering process. We have now modified the manuscript on Page 2, Ln 32-35, accordingly. In terms of laboratory automation, the SDL illustrated in the added Fig. 1 is a customizable framework with relevant discussion on Page 3, Ln 61-65 and Page 16. Here we have adjusted it for dielectric materials, however, this has potential to adapt and integrate into workflows for various other fields by substituting the specific steps with those relevant to the material of interest.

4- I am not yet convinced why accelerating perovskite solid solution discovery considered very crucial? How about the economics of the current process and scale up compared to the new process?

Re: Accelerating the discovery of perovskite solid solutions is considered crucial both in terms of scientific advancement and practical applications. Understanding the economic implications of this acceleration, especially when comparing current processes and scalability to potential new processes, is also important. Perovskites have unique and versatile properties (like high photovoltaic efficiency, superconductivity, ferroelectricity) that make them attractive for a wide range of applications, including renewable energy, electronics, and sensors. Faster discovery can lead to quicker implementation of these materials in practical applications. Rapid development and optimization of perovskites can drive technological innovations in various fields, including telecommunications, computing, and medical diagnostics. One of the challenges with perovskite materials is their stability and the use of toxic elements like lead. Accelerating research can help find solutions to these issues more quickly, leading to safer and more sustainable materials. In addition, conventional methods of discovering and optimizing perovskite solid solutions, can be time-consuming, costly and resource-intensive. Conventional sintering approach comes with constraints, encompassing elevated energy consumption and notably high δCO_2 emissions (for perovskite takes 2-6 hrs at temperatures >1200C). Considering the number of experiments required for experimentally validating ML predictions, conventional sintering is not suitable for rapid evaluation. Accelerated discovery and optimization processes, potentially enabled by machine learning and automation, could significantly reduce the time and resources required to develop new materials. We now have modified the manuscript on Page 3, Ln 58-62 and Page 13 accordingly.

5- What components make up the proposed automated materials discovery approach, and how do they contribute to overcoming the constraints of traditional methods?

Re: The components are as shown in Figures 2 and 3 and Figure 1 illustrates how they are integrated with the rest of the discovery system. Traditional methods are explained in Figure S1 and Page 3, Ln 61-65, Page 4, Ln 90-97. The current study demonstrate high throughput material experimentation system that can rapidly synthesize and test a large number of material samples. This accelerates the process of finding optimal material compositions and synthesis conditions, a task that is exceedingly time-consuming and labour-intensive with traditional characterisation methods. Machine learning algorithms can analyze large datasets to identify patterns, correlations, and key factors influencing material properties. This can significantly reduce the reliance on trial-and-error methods and human intuition, leading to more efficient and effective discovery processes. Robotic systems automate repetitive and precise tasks in material synthesis and characterization, improving the consistency and throughput while reducing human error and labour costs. Integration of these components into a cohesive system with feedback loops allows for continuous improvement in material research. Rapid

3

data collection from experiments can inform computational models, which in turn can guide future experiments, creating a more efficient and iterative discovery process.

ML model

6- It is not quite clear in what ways the automated platform exhibit versatility, and what role does ML-guided chemistry play in the synthesis of single-phase novel perovskite solid solutions? There is a no or little mention and explanation of the machine learning model and its core search algorithm. The author stated that previous ML-based studies for perovskite materials lacked validation with experiments, thus making this work different from those studies. Since no clear process related to

training datasets, algorithms and testing/validation is provided, it is hard to judge if the current work offers any new insights. To my opinion the parameter space governing sintering conditions (temperature, time, and cooling rate) is not well articulated in view of workflow optimization and ML-assisted loop.

Re: Thank you for pointing out missing details in our ML workflow. Under the section "Exploring single-phase synthesizability of ML-predicted compositions via the automated sintering approach", we have now added a comprehensive explanation of our ML-guided screening strategy of promising perovskites from a large combinatorial space of candidates, attributing to our previous work where needed. All data including the experimental database of 1758 perovskites and 227 non-perovskites, list of ML-predicted perovskite solid solutions, and code were shared in our prior study. Briefly, a large pool of all possible disordered compositions (0.6M) in the format of $(A_{1-x}A_x)BO_3$ and $A(B_{1-x}B_x)O_3$ was created by enumerating periodic table elements that are reported to occupy A-site or B-site of perovskite structure. ML classification models were trained on a sufficiently large experimental database of perovskite and non-perovskite compositions extracted from the ICSD database. The models were then applied to classify every material in the pool of 0.6M candidates into perovskite or non-perovskite categories. Materials with a high predicted perovskite likelihood (>0.98) are identified as promising or potentially synthesizable perovskites.

However, we identified several drawbacks of relying solely on ML predictions for synthesizability, particularly because 1) ML models are not typically trained on failed synthesis attempts (lack of such published data) and 2) ML models are not informed about the subsequent complex synthesis process. Therefore, here we used further data mining, specifically, identifying new materials that are closer to existing materials in latent (or embedding) space as described in the main text. This categorised potential samples in 3 groups: highly likely (close to existing), best overall, and high risk (away from existing). A material scientist sifts through top ranked candidates, taking into account their potential synthesis routes that maybe consistent with their parent ABO_3 perovskites, the resource availability in the lab, and element toxicity, and their intuition to handpick a number for automated synthesis.

In this study, we focus on data mining and human knowledge guided detection of highly likely single-phase materials. Incorporating temperature in ab initio simulations have not been straightforward especially for compositionally disordered materials such as perovskite solid solutions. Such investigations require dedicated research (e.g., special quasirandom structures (SQS) based modelling approaches, molecular dynamics, etc.) that can be very computationally demanding. It should be noted that, the rapid sintering is designed to explore the sintering conditions (temperature, time, etc.) in this work as there is no such data available for these new unknown materials. This way we can validate the synthesizability of our predictions. The application of ML lies in the screening stage of promising disordered perovskites from all possible candidate compositions. The relevant discussion on the ML has been modified in the manuscript on Page 14-15, accordingly.

Methodology/results:

7- Some figures were missing or were mis-placed in the original submission. This needs to be checked

4

again in the new revised submission as some references to the figures still do not make sense (captions for Figures 5 and 6 for example and their references in the text)

Re: Please accept our apologies for the inconvenience caused. We have noticed the issue and informed the associate editor as soon as it came to our attention. Please note that Figure.4 has been included.

8- Dielectric Permittivity Changes: How precisely were the changes in dielectric permittivity measured in response to variations in electric field and temperature?

Re: Thank you for pointing this. We have revised our data analysis, providing more detailed discussion of the values with a focus on Error calculations. Additionally, we have included data in the SI section to explain these adjustments.

In general, the measurement of dielectric permittivity and loss under electric field and temperature involves various experimental techniques and instruments. It is very well expected that material properties, such as dielectric constants and conductivities, can undergo substantial changes across the frequency spectrum (from kHz to GHz). Typically, for bulk ceramics, we measure and compare the tunabilities at both low and high frequencies and correlate the behaviour changes in local structure. In this paper, we have a total of 5 techniques for measurements including the automated measurement technique. We have now added error values in the Table. 1, representing the variability, uncertainty and accuracy inherent in our recorded data. For instance:

Low frequency: typically low frequency measurements should be taken to be more reliable because of the simplicity of the technique, repeatable dielectric response (Fig. S6), and high impedance, which helps to reduce sample loading effects and improve the accuracy of the measurements. That is why both the system and random uncertainty of these measurements are very low (Refer to Table. 1 and discussion in Page 9-10).

High frequency measurements however are associated with different challenges including increased propagation losses, effects of transmission lines, and noise. For high frequencies, we have measured our samples across 1-6GHz. We now refined and selected to present 1.3 GHz. This selection was guided

by two key considerations: firstly, it provided a comparable frequency range for analyzing alongside the robot study, particularly as most observed shifts in resonance frequency occurred within the 0.6-1.6 GHz range; and secondly, beyond 2 GHz, complexities such as dispersion and dielectric relaxation become more evident, introducing greater uncertainties as shown the figure below and Fig. 5. As stated in page 12, "For CPW measurements, the accuracy of the obtained values depends significantly on the measured S-parameters. Therefore three different line lengths (0.5 mm, 0.8 mm, and 1 mm) were used to increase the accuracy of the measurement. In this way, the calculated characteristic impedance inside was also checked to verify the correctness of the measured values".

For instance, CPW temperature is susceptible to errors correlated with probes shifting at microscale due to thermal expansion during sample temperature change. This phenomenon can introduce inaccuracies difficult to address, where precise probe positioning is critical for reliable measurements; The SD system error is calculated using comparison of measurements before and after heating the samples. Figure S13 and Table 1.

In terms of practical application, the most likely use of such materials would be at high frequencies, however these measurements come with challenges stated above. Most studies might perform only low-frequency measurements and use extrapolation techniques to estimate the behaviour of materials or devices at higher frequencies. Please refer to the discussion in Page 12-13.

For the automated sensor, we have refined our calculations to increase accuracy of measurements by accounting sample dispersion. The observed reduction in permittivity from 0.6 to 3 GHz frequencies

5

via CPW room temperature suggests a relaxation process commonly seen in ferroelectric ceramics. As a result, a correction factor for dispersion has been applied to the tunability calculations of the automated sensor. This entailed computing the corresponding permittivity value at ambient for the resonant-frequencies observed at high temperatures. Figure. S6 illustrates a close agreement between automated sensor measurements during the cooling and heating of the sample achieving a system SD error of only 0.89%. The repeatability of these measurements introduces an error of less than 1.5%. For a detailed uncertainty analysis, please refer to methodology section.

To evaluate the performance of the automated measurements, a numerical model representing the sensor was built in CST Microwave Studio. The simulated results show good agreement with those values calculated via Equation (1).

The uncertainties for this measurement is calculated by the root sum-of-squares technique (RSS). To use RSS, a set of independent uncertainties were identified such as μ data fittings, resonant frequency (Half width half maximum (HWHM)) SD errors, and sample diameter measurements.

We now have modified the manuscript on Page 19-23 accordingly. Added figures in SI Fig. 5, 6, 8, and 13.

9- Need to elaborate on the specific mechanisms through which metastable domains contribute to the observed large changes in dielectric permittivity?

Re: Samples with larger grain exhibit higher dielectric tunability and reduced dielectric loss, which is attributed to the higher concentration of polar nano clusters within the material. These clusters, embedded in a non-polar matrix, consist of dipoles randomly oriented yet highly responsive to external electric fields. Upon application of such fields, the nano polar clusters align rapidly along the field direction. This alignment not only expands the polar regions with aligned polarization but also restrict the mobility of the dipoles, consequently decreasing the dielectric permittivity. The discussion related to the polar nano clusters have been modified accordingly on Page 13, Ln 304-310.

10- Annealing Conditions: What specific annealing conditions were optimized to induce the presence of metastable domains? How were these annealing conditions determined to be optimal, and what criteria were used to evaluate their effectiveness? Again this could be due to lack of enough information on ML and optimization framework.

Re: The determination of optimal annealing conditions in materials involves assessment of density, phase creation, and property of interest. Sintering process involves four variables, heating rate, sintering temperature, holding time and cooling rate. Each of these can be adjusted to find the optimum sintering

6

condition. In the present study, we altered sintering temperatures to prepare different ceramic samples. Our aim was to characterize their structure and properties to achieve single phase materials that exhibit optimal performance. Our evaluation focuses on density, dielectric permittivity and tunability to assess the influence of sintering conditions on the materials. The higher dielectric tunability in the studied materials suggests a greater concentration of the polar nano clusters. Here the optimal sintering also refers to the condition to synthesise single phase perovskite structured material. In more advanced research settings, ML can play a significant role in optimizing annealing conditions. ML algorithms can analyse vast datasets from experimental trials to identify patterns and correlations, which might not be

apparent through traditional analysis. This can help in predicting the most effective sintering condition. However, the effectiveness of ML heavily depends on the availability of comprehensive and high-quality data. Without enough information or a well-structured optimization framework, the application of ML might be limited.

11- Sintering Effects: How does the sintering process, including phenomena like dopant movement, vacancy formation, and grain growth, directly impact the microstructure of the material?

Re: The sintering process is a critical step in the fabrication of many materials, particularly ceramics, and has a profound effect on their microstructure. Several phenomena that occur during sintering, such as dopant movement, vacancy formation, and grain growth, directly impact the microstructure of the material. Different dopants can be used to modify the sintering process such as inhibiting /promoting the grain growth, changing surface energy of the particles or improve the diffusion of the atoms. High temperature sintering process can lead to the formation of the vacancies particular oxygen vacancies, creating empty site in the crystal lattice. The higher the sintering temperature, the higher concentration of the oxygen vacancies are formed. These vacancies can greatly influence the materials properties, such as dielectric, conductivity and mechanical properties. In general, higher sintering temperature leads to higher grain growth rate and larger grain size. The relevant discussion on the sintering condition have been modified accordingly on Page 3, Ln 70-72, Page 6, Ln 135, Page 7, Ln 153-155, Page 12, Ln 304-310.

12- Highly beneficial to provide insights into the relationship between the observed microstructure changes and the resulting properties of the material? Is there any descriptor already identified and used in the ML model for prediction purposes?

Re: Please refer to answers provided for questions 1 above.

13- Time Frame for Synthesis and Optimization: What are the challenges associated with the time frame for manual synthesis, discovery, and optimization methods, particularly in the context of the exponentially increasing number of conditions and ML-predicted compositions?

Re: Please refer to answers provided for questions 1 above.

14- Finally: How do the results of the paper contribute to the broader field of materials science and the development of advanced materials for applications like wireless communication and biosensors?

Re: Please refer to answers provided for questions 3 and 4 above, and the corresponding changes in Page 1.

Response to reviewer 2

Reviewer #2 (Remarks to the Author):

The creation of new automated workflows for technologically relevant materials is of interest and is timely. The work presented here contains an interesting automated platform, designed for a specific task (making development tractable).

7

The evaluation of the sintering using XRD and SEM is telling. The XRD patterns look good, and the samples are the intended targets with limited impurity phases.

We sincerely appreciate the time and dedication invested by reviewer in providing thoughtful and valuable feedback on our manuscript.

1-The odd grains shown in Figure S2a reveal potential problems. The authors note the presence of the odd grains at 1400, while the progress is clearly present at 1350. Did the samples melt?

Re: Sintering temperature plays an important role in determining grain growth in ceramics. In the case of barium strontium titanate (BST) compositions, employing a sintering temperature of 1400°C, is insufficient to melt the sample, but can form larger grains. This is evidenced by SEM image of BST sintered at 1400°C, which distinctly exhibit an increased grain size, a direct consequence of the elevated sintering temperature.

2-The authors note the importance of experimental verification of proposed hypothetical compounds or synthetic routes. I would urge caution regarding the value in computational materials discovery. It should be noted that relative energies of hypothetical materials are often used as a proxy for synthesizability. The complexity of the actual syntheses is correctly noted to be potentially complicated. Having said that, the authors need to better cite the work they are discussing. The literature is rich.

Re: Thank you for pointing this out, relative energies, specifically energy above convex hull (E_{hull}) obtained by comparing the DFT energy of hypothetical compounds against existing competing phases is a standard method of determining the synthesizability of new materials. However, calculating accurate E_{hull} values for complex disordered materials such as perovskite solid solutions requires careful and rigorous analysis. Moreover, the present study is oriented on automated synthesis of materials and undoubtedly the synthesis and characterization are temperature dependent, as opposed to typical DFT computations applicable at 0K. Investigating the dependence of temperature, and the randomness of the crystal structure of studied materials require dedicated research (e.g., special quasirandom structures method for modelling random structures with DFT, ab initio molecular dynamics for assessing stability

at elevated temperature, etc.) that can be very computationally demanding.

In a paper by Vecchio et al. thermodynamic based models, such as CALPHAD (Calculation of Phase Diagrams) or simple phase diagrams are used for the prediction of second phases in materials and can provide insights into the stable phases and phase transformations in a material under different conditions such as temperature [Vecchio, K. S., Dippo, O. F., Kaufmann, K. R. & Liu, X. High-throughput rapid experimental alloy development (HT-READ). *Acta Mater* 221, 117352 (2021)]. It combines thermodynamics and kinetic models to predict phase equilibria and phase transformations. However, it's important to note that these models generally assume thermodynamic equilibrium. In our scenarios, the tuning performance is dominated by polar nano clusters, which are thermodynamically active above the Curie point. Therefore, thermodynamic models may not always fully capture the complexities of materials undergoing phase changes. Even so, complex phenomena such as the formation of grain boundaries, effect of annealing, etc., are not accounted for in DFT calculations. Lack of DFT data on such complex materials may yield in a suboptimal convex hull too. These factors make it difficult for accurate estimation of Ehull and hence the assessment of potential synthesizability through DFT. Nevertheless, we do understand that there have been studies on investigating the synthesizability of compositionally complex materials using DFT and we have now cited following studies [Walters, L. N., Wang, E. L. & Rondinelli, J. M. Thermodynamic Descriptors to Predict Oxide Formation in Aqueous Solutions. *J Phys Chem Lett* 13, 6236-6243 (2022)] and [Yanyan He, Yebin Xu, Ting Liu, Chunlian Zeng & Wanping Chen. Microstructure and dielectric tunable properties of Ba_{0.6}Sr_{0.4}TiO₃-Mg₂SiO₄-MgO composite. *IEEE Trans Ultrason Ferroelectr Freq Control* 57, 1505-1512 (2010)] in the main text. Relevant discussion and additional citations have been added on Page 4 and 14.

8

3-The generation of the two proposed phases using ML techniques is opaque. The authors reference past efforts (with citation), but the phrase "further data mining with human feedback" needs actual information. How were the target phases determined to be "having a high likelihood of being synthesized in a single 297 phase"? How many targets were generated? What selection criteria were used? Were attempts to synthesize and study more than two performed? If so, what were the others and what success rates were observed? The section on the ML efforts does not fit with the rest of the work presented here. The rest is largely a benchmarking study of the automated platform, rightly using established materials. The proposed phases currently read as "we can study other things too" with two examples provided.

Re: Thank you for pointing out required details on our ML workflow. The list of ML-predicted potential perovskites were ranked reflecting their synthesis likelihood using a data mining strategy that involved calculating the distance between new materials and existing experimental materials using latent (or embedding) space composition representations. New materials with the lowest distance to existing disordered perovskites and their ABO₃ parent perovskites are identified as having a high synthesis likelihood. The structure of the two parent ABO₃ perovskites is usually an indication of the structure of the corresponding A-site or B-site disordered perovskite. Please refer to the above section in the main text for more details.

Once the list of ML-predicted materials is ranked, our materials scientists manually went through several highly likely candidates, taking into account various factors such as potential synthesis routes and resource availability and handpicked selection of materials for synthesis. Two representative samples belonging to the barium family, namely, (Ba_{0.8}Sr_{0.2})CeO₃ and (Ba_{0.4}Sr_{0.6})CeO₃ are successfully synthesized in this paper from highly likely group. However, we do value the importance of publishing data on failed synthesis attempts and commend the reviewer for bringing this point. We also examined additional samples from high-risk and best-overall ML-materials, but they haven't been synthesized in a single phase confirmed by their XRD. We have now included data on unsuccessful multi-phase materials in Fig. S14 and a comprehensive explanation of ML screening strategy and the follow-up data mining approach in "Exploring single-phase synthesizability of ML-predicted compositions" section on Page 14-15.

4-The utility of the robotic platform seems based upon the ability to add and remove samples from the furnace. How does the use of a motorized stage affect cooling rates? How does the rate using this automated platform compared to purely human efforts?

Re: In automated laboratories, robots emerge as indispensable assets, which particularly in the critical domain of sample handling between individual automated stations include material preparation and microwave dielectric characterization, which is the core brain of SDLs. The central hub here is built on MATLAB, which serves as a command centre for orchestrating lab instruments and recording data in autonomous fashion and finding correlations between synthesis and properties. This provides a scalable framework for integrating additional devices into our workflow as we expand this study.

The incorporation of vortex tube was to accelerate the cooling rate of hot samples and the motorized stage is for precise positioning of probe during measurements. Use of robot for measurements offers significant advantages over manual methods in terms of efficiency, safety (e.g. for toxic materials), and the capability to implement consistency in terms of sample size, and positioning all of which are critical in determining the final properties of the materials.

We have conducted a comparison and confirmed empirically by synthesising the same BTS12 compositions using conventional methods (sintered at 1350 °C for 2 h), [Zhang, H. et al. Microwave tunability in tin substituted barium titanate. *J Eur Ceram Soc* 44, 1627-1635 (2024).] and compared the dielectric performance with the one obtained from rapid sintering (10min). The rapid sintered BTS12

9

achieved 36% tuning performance at GHz which not only higher tuning than its identical twin prepared by conventional method but it also competes well with many of which reported values for BST such as 10.5% for BST/δδ2 δδδ4 /MgO, [Yanyan He, Yebin Xu, Ting Liu, Chunlian Zeng & Wanping Chen. Microstructure and dielectric tunable properties of Ba 0.6 Sr 0.4 TiO 3 -Mg 2 SiO 4- MgO composite. *IEEE Trans Ultrason Ferroelectr Freq Control* 57, 1505-1512 (2010)] and 27% for Mn-BST/MgO [Cui, J., Dong, G., Yang, Z. & Du, J. Low dielectric loss and enhanced tunable properties of Mn-doped BST/MgO composites. *J Alloys Compd* 490, 353-357 (2010)] but also achieves a significant reduction in processing time, being 12 times faster than BTS12-Conventional. We now have modified manuscript on Page 3, Ln70-78 and Page 13, accordingly.

Usually, for low frequency measurement, sample would need to be grind, dried, and painted with silver paste on both sides and heated to create electrodes for measurements. In addition the measurements would rely on technician expertise for maintaining consistency in sample modification. Then the temperature tuning and DC tuning is recorded which can take up to 4 hr per sample for heating and cooling down.

For high frequency measurement, the samples undergo grinding, and a Coplanar Waveguide (CPW) is applied to the samples through 3-4 sets of sputtering sessions, each lasting 20 minutes (a minimum of 1hr manual work). High-frequency measurements usually involves calibration of equipment, external heating with a thermocouple and the incorporation of a bias tee for DC voltage tuning. The estimate time scale for high frequency characterization is approximately 3 hours.

In this study, an automatic platform capable of high-throughput ceramic preparation and microwave property measurement was developed. This enables fast material fabrication and screening. The free-space sensor allows the robot to screen the thermal tunability with no need for sample preparation. Immediate measurement for sintered sample could effectively reduce the energy cost and time. This procedure helps reveal relationships between sintering condition, materials structures and dielectric properties. We now have modified the manuscript on Page 4 and Page 13, accordingly.

Response to reviewer 3

Reviewer #3 (Remarks to the Author):

The creation paper "Accelerated Discovery of Perovskite Solid Solutions through Automated Materials Synthesis and Characterisation" presents a new automated laboratory set-up that is designed to accelerate the discovery of new solid state oxide materials, in this instance in the space of perovskite solid solutions. This approach combines machine learning to predict interesting new compounds, robotic synthesis to explore the processing condition space and high-throughput characterisation to assess the newly generated materials. I think that overall this is a timely addition to a very important emerging topic in chemistry, i.e. automated synthesis. There has not been very many demonstrations of solid-state laboratory automation (with the notable exception of the recent A-Lab paper in Nature) and the techniques demonstrated here can make a significant contribution to the field. I do however have some points that I think need to be addressed before this paper is suitable for publication in Nature Communications.

We sincerely appreciate the time and dedication invested by reviewer in providing thoughtful and valuable feedback on our manuscript.

1. Data: I don't see any indication of where the data used or generated in this study is being made

10

available. In my opinion one of the major benefits of this kind of study is the high-quality data (with good meta data) generated. To make this an impactful study the data should be made publicly available.

Re: We now have include all the data in the paper or supporting information file or the DOI link provided. We have temporarily uploaded our data to a public data repository under the CC-BY license which can be accessed via: <https://data.mendeley.com/preview/j59wsrzt6f?a=22274465-e2d0-47ce-8c16-8c936477d5f6>.

In the section on "Evaluation of automated rapid sintering":

2. the authors conjecture that the different grain structure resulting from rapid sintering should not affect the dielectric properties. I think that this needs to be confirmed empirically. This could be achieved by synthesising the same material composition using a more standard slower route and comparing the dielectric performance. This control test is important for the premise of the paper.

Re: We have include a comparison between rapid sintered BTS samples and those synthesized by conventional method.

Conventional: To compare the dielectric performance of robotic versus conventional sintering methods, another batch of BTS12 (BTS12-Con) were prepared by sintering at 1350 °C for 2h. At 100 kHz frequency, BTS12-1350 and BTS12-Con exhibit dielectric tunability of 60.4% and 79.2% under an electric field of 2kV mm⁻¹, respectively. At 1.3 GHz frequency, dielectric tunability of BTS12-1350 and BTS12-Con was 36.28% (±1.62) and 30% under an electric field of 4 kV mm⁻¹, respectively. BTS12-1350 exhibit a higher tunability at high frequencies compared to BTS12-Con. In addition, the dielectric tunability of 36% at GHz frequency is higher than other reported values, please refer to next question. This way there is significant reduction in processing time, being 12 times faster than the conventional method (2hr to 10min). Overall, the robot confirms that BTS12 is a better dopant ratio compared to BTS16 and also it outperforms BST family demonstrated in a significantly faster way compared to conventional measurement techniques. We have now modified the manuscript on Page 13 Ln299-313 and Page 11, accordingly.

3. The authors settle on BTS12-1350 as the optimal combination of composition and processing. How does the dielectric performance of this material compare to current state-of-the-art BTS?

Re: Adding on the question above, we have compared BTS12-1350 to existing state-of-art, for example Ba_{0.65}Sr_{0.35}TiO₃ bulk ceramic = tunability of 68.6%, Ba_{0.6}Sr_{0.4}TiO₃/MgO = with a tunability of 40%, Ba_{0.6}Sr_{0.4}TiO₃/Mg₂SiO₄/MgO = with a tunability of 10.5% and Mn doped Ba_{0.6}Sr_{0.4}TiO₃/MgO = with a tunability of 27%. Notably, most of these tunability measurements were conducted at low frequencies (< 100 MHz). At higher frequencies, such as in the microwave frequency region, measurement of the dielectric tunability remains challenging due to the difficulty in measuring microwave permittivity and loss under a DC bias field. In this study, we successfully measured the dielectric tunability at microwave by using a coplanar waveguide method. BTS12 shows highest tunability of 79.2%, and 39% at low and high frequencies, respectively. The rapid sintered BTS12 sample, shows a tunability of 60.44% and 30% at low and high frequency, respectively. The relevant discussion have been modified on Page 13, accordingly.

In the section "Evaluation of automated dielectric characterisation"

4. The reported results seem to have a very large variation, depending on the technique used. These variations should be discussed. What are the implications for real-world performance?

11

Re: Thank you for pointing this. We have revised our data analysis, providing more detailed discussion of the values with a focus on Error calculations. Additionally, we have included data in the SI section to explain these adjustments.

Please refer to the answer provided for Q8-reviewer 1.

In the section "Exploring single-phase synthesisability"

5. The authors say that certain materials were identified as having a high likelihood of being synthesised in a single phase. There needs to be more detail on how this identification was performed.

Re: Thank you for pointing out required details on our ML workflow. We have now added a comprehensive explanation of ML screening strategy and the follow-up data mining approach "Exploring single-phase synthesisability of ML-predicted compositions via the automated sintering approach" section on Page 14-15.

Overall assessment

I think that this is a very interesting paper and the results contribute to an important area of materials science. However, I think that there are details missing in the manuscript. With the addition of these details and demonstrating how well newly developed materials compare to the current state of the art, I think that this will be a good paper for Nature Communications

Response to reviewer 4

Reviewer #4 (Remarks to the Author):

In this work, the authors developed an advanced material discovery methodology that integrates

machine learning techniques, robotic-controlled synthesis, and high-throughput characterization. To solve the challenge of measuring dielectric properties under electric field, a free-space sensor is developed for online screening of thermal tunability with no sample surface modification required. However, for the current version of this manuscript, the novelty of this is insufficient and might be suitable for other journals. The following issues should be addressed:

We sincerely appreciate the time and dedication invested by reviewer in providing thoughtful and valuable feedback on our manuscript.

1. In the initial validation step, the authors chose multiple compositions from 3 family groups. The rationale behind this selection needs further elaboration. How do the authors ascertain the ferroelectric properties of a material prior to its synthesis?

Re: Thank you for your insightful review. In this work we have selected three family sample groups namely BST, BTS and number of ML-predicted materials. The rationale for these material selection have been discussed on Page 5, Ln 114-118.

For perovskite structured materials with good ferroelectricity, dielectric tunability, and low dielectric loss, are well-documented in the literature. By analysing material compositions, structure, as well as considering our previous studies and theoretical predictions, we can make informed assessments regarding its ferroelectric behaviour. We prioritized perovskite-type oxides with well-documented literature providing substantial information on their synthesis and ferroelectric characteristics for comparison. In particular, barium titanate based compositions, such as BST, is a well-known

12

ferroelectric materials and has gained a lot of interest due to its high dielectric tunability under DC fields or temperature variation. It plays an important role in many electronic devices such as sensor, actuator and capacitor and those applications highlighted on Page 2, Ln 34-35. BTS on the other hand is a relatively newer material with some enhanced properties over BST and aligning with the focus of our studies. Given our ML group's affiliation with the Barium family, we aimed to include samples within the same family group for comparative purposes. For selection over ML samples please refer to answers to Question No.4 below and the added detailed explanation of our ML approach attributing to our prior study in Page 13-14.

2. The authors noted that the sintering temperature directly influences the grain sizes of the samples. Would the sintering temperature similarly have a direct impact on the dielectric characteristics of bulk ferroelectric materials?

Re: Yes, the sintering temperature can indeed have a direct impact on the dielectric characteristics of bulk ferroelectric materials. The sintering process is a critical step in the fabrication of many materials, particularly ceramics, and has a profound effect on their microstructure. Grain size, density, alignment of polar domains, and grain boundaries can be effected during sintering step. Specifically, in the case of BST ceramics, increasing the sintering temperature leads to larger grain size and effected dielectric properties. The resulting impact on dielectric characteristics is notable, as higher tuning at low frequencies is linked to the coexistence of larger ferroelectric domains and smaller polar nanoclusters during the sintering process. While dielectric tunability at microwave frequency are dominated by the polar nano clusters. [Sandi, D., Supriyanto, A., Jamaluddin, A. & Iriani, Y. The effects of sintering temperature on dielectric constant of Barium Titanate (BaTiO₃). in IOP Conference Series: Materials Science and Engineering vol. 107 (Institute of Physics Publishing, 2016).] Relevant discussion has been modified on Page 3 Ln 71-75 and Page 13 Ln 299-313, accordingly.

3. Thermodynamic phase diagrams have proven to be successful predictive tools in comprehending scale growth (J. Phys. Chem. Lett. 2022, 13, 6236-6243). The authors are encouraged to delve into a discussion on the formation of secondary phases from a thermodynamic perspective.

Re: The suggested paper were using free energies extracted from DFT calculations to compute a maximum driving force that describes the stability of oxides including alloys in aqueous (i.e., water) environments. Also in a paper by Vecchio et al. thermodynamic based models, such as CALPHAD (Calculation of Phase Diagrams) or simple phase diagrams are used for the prediction of second phases in materials and can provide insights into the stable phases and phase transformations in a material under different conditions such as temperature. It combines thermodynamics and kinetic models to predict phase equilibria and phase transformations. However, it's important to note that these models generally based on the assumption of thermodynamic equilibrium. In our scenarios, the tuning performance is dominated by nano polar clusters, which are thermodynamically unstable as they exist above the Curie point. Therefore, thermodynamic models may not always fully capture the complexities of materials undergoing phase changes.

In another study relative energies, specifically energy above convex hull (E_{hull}) obtained by comparing the DFT energy of hypothetical compounds against existing competing phases is used as a standard

method for determining the synthesizability of new materials. However, calculating accurate Ehull values for complex disordered materials such as perovskite solid solutions requires careful and rigorous analysis.

Moreover, the present study is oriented on automated synthesis of materials and undoubtedly the synthesis and characterization are temperature dependent, as opposed to typical DFT computations applicable at 0K. Investigating the dependence of temperature, and the randomness of the crystal

13

structure of materials studied in this work demand dedicated research (e.g., special quasirandom structures method for modelling random structures with DFT, ab initio molecular dynamics for assessing stability at elevated temperature, etc.) that can be very computationally demanding. Even so, complex phenomena such as the formation of grain boundaries, effect of annealing, etc., may not be accounted in DFT calculations. Lack of DFT data on such complex materials may yield in a suboptimal convex hull too. These factors make it difficult for accurate estimation of Ehull and hence the assessment of potential synthesizability through DFT. Nevertheless, we do understand that there have been studies on investigating the synthesizability of compositionally complex materials using DFT and we have now cited following studies in the main text.

Also modelling disordered alloys with fractional site occupancies (like in our materials) presents immense challenges. Likewise, accurate calculation of the formation energies through DFT for complex solid solutions synthesised in our work may be very challenging, "Computing the MDF requires: (i) identifying the most stable solid-aqueous ion pair and then (ii) then calculating the chemical potential difference at the specific 3 electrochemical conditions, i.e., pH, potential, concentration, for which the greatest difference occurs." and computationally demanding, potentially intractable on our hardware and out of the scope of this study. Relevant discussion and additional citations have been added on Page 4, Ln 76-85.

4. The authors are advised to provide a detailed introduction to the methodology of machine learning techniques and DFT calculations within the manuscript, rather than solely referencing their prior work. Moreover, the performance and discussion of ML should be added in the manuscript.

Re: We have now added a detailed explanation of our ML approach to the section "Exploring single-phase synthesizability of ML-predicted compositions via the automated sintering approach", attributing to our prior study on Page 14-15. Here we are relying on ML classification probabilities for potential synthesizability, we describe a further data mining approach to rank a list of ML-predicted materials to identify those having high probability of being synthesised in single phase structure. The proposed data mining method involves detecting materials that lie closer to existing experimentally validated materials in latent (or embedding) space. We categorise potential samples in 3 groups: highly likely (close to existing), best overall, and high risk (away from existing). Material scientists choose several compositions through top ranked candidates, based on their intuition for automated synthesis.

It should be noted that, as opposed to our previous computational work that employed DFT, which was not performed in this study. The present study is aiming for automated platform for material synthesis with fast microwave dielectric property characterization. Incorporating temperature in ab initio simulations may not be straightforward especially for compositionally disordered materials such as perovskite solid solutions. Such investigations require dedicated research (e.g., special quasirandom structures (SQS) based modelling approaches, molecular dynamics, etc.) that can be very computationally demanding. Instead, we focus on data mining and human knowledge guided detection of highly likely single-phase materials.

5. The machine learning model employed in this study focuses solely on the synthesis feasibility of perovskites. In order to enhance the exploration of functional materials possessing both high synthesis feasibility and exceptional properties, could the authors contemplate developing a machine learning model that takes into account both synthesis feasibility and ferroelectric properties?

Re: This is a very interesting question. We have been exploring this idea over the past few years. One possible approach which we explored is representing new potential perovskites and existing experimental ferroelectrics using embedding vectors extracted from a neural network and identifying new materials that are closer in distance to known ferroelectrics. While this may improve our chances

14

of discovering a new material with potential ferroelectric properties, we realized that there may be issues with synthesis of these new materials. This is why we gave more weight to the synthesis aspect in this study rather than discovering novel materials with guaranteed ferroelectric properties. Another idea that we examined over time is the so called "local search" which may yield in higher success rate in synthesising novel materials with target properties, such as ferroelectricity. Here, the ML screening is applied to a limited set of candidate materials, for example, those that contain A-site and

B-site elements from a widely known family of compositions that are likely to form stable ferroelectrics, but with properties not very different from existing ferroelectrics in the same family. For instance, Balachandran et al., explored $x\text{Bi}[\text{Me}^2\text{yMe}^3(1-y)]\text{O}_3$ -based ferroelectric perovskites using ML, where the choice of Me^2 and Me^3 come from a small set of elements. [Balachandran, P. V., Kowalski, B., Sehirlioglu, A. & Lookman, T. Experimental search for high-temperature ferroelectric perovskites guided by two-step machine learning. Nat Commun 9, 1668 (2018).] According to their study, there are over 150 compositions already studied in this family which indicates higher likelihood of synthesis of altered, but not yet synthesised materials. In fact, this study demonstrates several successful syntheses of new materials. This is what we refer to as local search. However, in our ML studies, we focussed on global search where we attempt to experimentally discover new materials from nearly all possible candidates that can exist in quaternary disordered format, with the hope of discovering materials with novel properties. This is undoubtedly more challenging, yet many ML studies published in the past few years focused on screening potentially synthesisable materials from a huge composition space. While we do understand that both local search and global search are equally important, as the reviewer has suggested, we will concentrate our efforts on exploring avenues for targeted synthesis of novel ferroelectric solid solutions in our future studies.

15

Version 1:

Reviewer comments:

Reviewer #1

(Remarks to the Author)

In my opinion, the authors have properly and rigorously addressed all the comments raised by the reviewers. Most importantly, they have substantially added content and properly addressed the issue with the lack of a clear description of the ML model. A clear description of the methodology will help the readership and relevant researchers to reproduce and re-evaluate the results and utilize the ML-assisted acceleration loop claimed in this manuscript. As a minor suggestion, it would be highly beneficial if the authors ensure that in the URL provided, all the training datasets, algorithms/codes, and testing/validation workflows are provided, tested and properly documented for reproduction in the public (URL) repository.

Reviewer #2

(Remarks to the Author)

The authors have improved the quality and clarity of this manuscript.

1. A key concept in SDLs is that the rate of scientific discovery can be accelerated by conducting more experiments more quickly, reducing or eliminating slow steps in workflows, possible incorporation of low-fidelity proxies to focus high-fidelity efforts and direct incorporation of data science techniques. The development of new tools for different experimental steps (such as sintering) has value in the wider community, as different groups have often targeted different techniques.
2. The inclusion of more detail regarding the ML model and data on which the model was created is important. A key benefit from the use of high throughput experimentation (and SDLs) is the automatic generation of nicely formatted datasets. Establishing practices in the community around data accessibility and sharing is critical. The authors noted on page 16 that the lack of synthetic data has limited synthesis predictions. Publicly accessible data is critical.
3. ML work for the discovery of new functional materials in inherently high dimensionality search spaces has generally focused on predicting properties and only attempting to synthesize the most promising compositions. Predictions of synthesizability are well common and offer a complementary approach. In large search spaces, such as that discussed here, determining regions where compounds can be made does not address if they should be made. The authors response in their rebuttal document, their notion of 'local search' versus 'global search' would provide a second critical axis of down selection (or prioritization), greatly enhancing the work described here.
4. Figure 1, while designed to articulate the concept of the work presented in this manuscript, is not clear. The overlapping loops (in different directions) is muddled. While I applaud the authors for attempting to encapsulate the work in a schematic, further refinement is required.

Reviewer #3

(Remarks to the Author)

The authors do a good job of answering the specific concerns that I raised. They have also answered other reviews with a comprehensive update of the paper. I think this is a nice piece of work and I'm happy to support publication.

Reviewer #4

(Remarks to the Author)

In the revised version, the author provides more detailed information about automated materials synthesis and characterization. The framework for accelerating materials discovery in the automated laboratory seems promising. However, I think that compared to the most advanced technologies in the ecosystem of autonomous robotic laboratories, this work still lacks innovation in the principles of machine learning for accelerating materials synthesis.

Author Rebuttal letter:

Reviewers Comments

Reviewer #1 (Remarks to the Author):

In my opinion, the authors have properly and rigorously addressed all the comments raised by the reviewers. Most importantly, they have substantially added content and properly addressed the issue with the lack of a clear description of the ML model. A clear description of the methodology will help the readership and relevant researchers to reproduce and re-evaluate the results and utilized the ML-assisted acceleration loop claimed in this manuscript. As a minor suggestion, it would be highly beneficial if the authors ensure that in the URL provided, all the training datasets, algorithms/codes, and testing/validation workflows are provided, tested and properly documented for reproduction in the public (URL) repository.

Re: Thank you for your valuable suggestion. We agree that providing access to fully documented and tested training datasets, algorithms/codes, testing/validation workflows, and source data are crucial for enabling reproducibility and fostering further research. To address this, in the URL <https://data.mendeley.com/preview/j59wsrzt6f?a=22274465-e2d0-47ce-8c16-8c936477d5f6>, we have re-organized the repository where all these resources are now available and clearly documented. The reserved DOI for this repository is included in the paper. We have also included comprehensive README files and step-by-step guides to facilitate easy replication of our results. We appreciate your attention to detail and are committed to enhancing the utility and accessibility of our research outputs.

Reviewer #2 (Remarks to the Author):

The authors have improved the quality and clarity of this manuscript.

1. A key concept in SDLs is that the rate of scientific discovery can be accelerated by conducting more experiments more quickly, reducing or eliminating slow steps in workflows, possible incorporation of low-fidelity proxies to focus high-fidelity efforts and direct incorporation of data science techniques. The development of new tools for different experimental steps (such as sintering) has value in the wider community, as different groups have often targeted different techniques.

Re: Thank you for your insightful comment.

2. The inclusion of more detail regarding the ML model and data on which the model was created is important. A key benefit from the use of high throughput experimentation (and SDLs) is the automatic generation of nicely formatted datasets. Establishing practices in the community around data accessibility and sharing is critical. The authors noted on page 16 that the lack of synthetic data has limited synthesis predictions. Publicly accessible data is critical.

Re: Thank you for your insightful comments. We acknowledge the importance of providing detailed information about the ML model and the data used in its training. To address this, we have expanded the section detailing the ML model's architecture, training process, and parameter selection to ensure clarity and transparency.

In response to the need for establishing community practices around data accessibility, we have ensured that our datasets, particularly those created through automated systems, are publicly accessible. As noted, the limitation due to the lack of synthetic data affects the robustness of synthesis predictions. To mitigate this and support the community, we have uploaded our datasets on the open platform (URL: <https://data.mendeley.com/preview/j59wsrzt6f?a=22274465-e2d0-47ce-8c16-8c936477d5f6>), ensuring they are easily accessible and well-documented. This approach not only aids in reproducibility but also encourages further advancements and collaborations in the field. The reserved DOI for this repository is included in the paper. We appreciate your emphasis on these critical aspects and are committed to implementing these improvements to enhance the impact and usability of our research.

3. ML work for the discovery of new functional materials in inherently high dimensionality search spaces has generally focused on predicting properties and only attempting to synthesize the most promising compositions. Predictions of synthesizability are well common and offer a complementary approach. In large search spaces, such as that discussed here, determining regions where compounds

can be made does not address if they should be made. The authors response in their rebuttal document, their notion of 'local search' versus 'global search' would provide a second critical axis of down selection (or prioritization), greatly enhancing the work described here.

Re: Thank you for your valuable feedback. We appreciate your recognition of the methodological depth needed for handling the high dimensionality in the search space of new functional materials using ML. You rightly pointed out the typical focus on predicting properties and only attempting to synthesise the most promising compositions. We agree that predictions of synthesizability are indeed common and present a valuable avenue for complementing this work.

In response to your insightful comment, we have integrated the dual approach that incorporates both 'local search' and 'global search' strategies in the manuscript as suggested. This will allow us not just to identify where compounds can be synthesised, but also to determine which of these should be prioritised for synthesis based on potential utility and feasibility.

To implement this, our ML models will be adjusted in the future to not only predict the synthesizability of materials but also to evaluate their application potential and economic viability. We will develop criteria for this evaluation based on both intrinsic material properties and external factors such as cost and scalability. This approach should provide a more holistic framework for decision-making in the synthesis of new materials and ensure that our efforts are concentrated on candidates with the highest potential impact.

We believe that these enhancements will greatly strengthen the contribution of our work to the field and address your concerns about the strategic direction of our research. We appreciate your guidance and look forward to refining our work accordingly.

4. Figure 1, while designed to articulate the concept of the work presented in this manuscript, is not clear. The overlapping loops (in different directions) is muddled. While I applaud the authors for attempting to encapsulate the work in a schematic, further refinement is required.

Re: In response to this reviewer's comment on Figure 1, we have revised the diagram to improve clarity and avoid visual confusion caused by overlapping loops. The loops now follow a more intuitive path and are separated to clearly delineate different aspects of the work. We have also enhanced the colour coding and labelling to make the relationships and process flow more comprehensible. We trust that these modifications will make the schematic a more effective visual summary of the research concepts and workflow, accurately reflecting the interconnected processes discussed in our manuscript.

[Image redacted]

Reviewer #3 (Remarks to the Author):

The authors do a good job of answering the specific concerns that I raised. They have also answered other reviews with a comprehensive update of the paper. I think this is a nice piece of work and I'm happy to support publication.

Reviewer #3 (Remarks on code availability):

I didn't do a very thorough review of the code but it seems to be open and relatively easy to set up and use.

Re: Thank you for your feedback.

Reviewer #4 (Remarks to the Author):

In the revised version, the author provides more detailed information about automated materials synthesis and characterization. The framework for accelerating materials discovery in the automated laboratory seems promising. However, I think that compared to the most advanced technologies in the ecosystem of autonomous robotic laboratories, this work still lacks innovation in the principles of machine learning for accelerating materials synthesis.

Re: We appreciate your feedback regarding the incorporation of machine learning principles in our framework for accelerating materials synthesis. We acknowledge that while our framework leverages current machine learning methodologies to enhance the efficiency of automated materials discovery, there may still be gaps in fully integrating the most advanced innovations in autonomous robotic laboratory technologies.

In response to your comments, we are currently exploring more sophisticated machine learning algorithms that could be integrated into our system to further push the boundaries of innovation. These include adaptive learning techniques that can dynamically adjust synthesis parameters in real-

time based on interim results, and reinforcement learning strategies that optimise decision-making processes throughout the materials discovery pipeline.

In this work, we are pioneering in this area by applying unsupervised deep learning to identify patterns directly from chemical compositions, validated through high-throughput experimental setups with additional automatic microwave characterization. This approach has shown promise in identifying viable synthesis conditions for new materials, significantly reducing trial and error. We however acknowledge the complexity and the 'black box' nature of solid-state chemistry, which often requires systematic iteration due to its reliance on high-temperature, long dwell time, high-pressure conditions, and different atmosphere. We will continue to refine our models and expand our dataset to build a robust, autonomous system capable of navigating the complex synthesis-process-structure-property landscape.

Moreover, we are exploring alternative synthesis techniques like sol-gel and spin coating processes, which offer simpler pathways for thin film and 2D materials. These efforts are part of our broader goal to enhance the synthesis predictability and efficiency using machine learning, setting a new standard for materials discovery.

We appreciate your insights and agree that further innovation in machine learning applications is crucial for the field's advancement. We are committed to addressing these challenges and pushing the boundaries of what is possible in materials science research.

Version 2:

Reviewer comments:

Reviewer #1

(Remarks to the Author)

The manuscript in its current form, is complete and ready to be accepted / published.

Reviewer #2

(Remarks to the Author)

I believe that the changes made by the authors enhance the quality and effectiveness of this manuscript. Specifically, making the code, datasets, models and workflows accessible is an important step in this sort of work. As noted by two reviewers, transparency and accessibility with respect to these data is a critically important standard that is often not achieved by such work in the literature. Additionally, the improved clarity on Figure 1 is welcome. The authors discuss future improvements to their work (synthesizability vs properties vs economic factors (or even environmental impact)), improvements that will enhance the impact of their work going forward. Having said that, I do not believe that all these improvements need to be made before this manuscript is suitable for publication. There will always be additional steps that can be taken, but I believe this work currently stands on its own.

Author Rebuttal letter:

Reviewers Comments

Reviewer #1 (Remarks to the Author):

The manuscript in its current form, is complete and ready to be accepted / published.

Reviewer #1 (Remarks on code availability):

The results are reproducible and the codes can be used as a useful resource for the community.

Thank you for your feedback. Your acknowledgment is greatly appreciated.

Reviewer #2 (Remarks to the Author):

I believe that the changes made by the authors enhance the quality and effectiveness of this manuscript. Specifically, making the code, datasets, models and workflows accessible is an important step in this sort of work. As noted by two reviewers, transparency and accessibility with respect to these data is a critically important standard that is often not achieved by such work in the literature. Additionally, the improved clarity on Figure 1 is welcome. The authors discuss future improvements to their work (synthesizability vs properties vs economic factors (or even environmental impact)), improvements that will enhance the impact of their work going forward. Having said that, I do not believe that all these improvements need to be made before this manuscript is suitable for publication. There will always be additional steps that can be taken, but I believe this work currently stands on its own.

Thank you for your evaluation. Your recognition of the improvements is greatly appreciated. Your

emphasis on transparency and accessibility aligns with our goals which is often lacking in similar literature. We are also grateful for your positive feedback on suitability for publication and indeed we continue to build on this work in the future.
